# An exceptionally preserved *Sphenodon*-like sphenodontian reveals deep time conservation of the tuatara skeleton and ontogeny

Tiago R. Simões [1]✉, Grace Kinney-Broderick[2,3] & Stephanie E. Pierce [1]✉

Sphenodontian reptiles are an extremely old evolutionary lineage forming the closest relatives to squamates (lizards and snakes) and were globally distributed and more diverse than squamates during the first half of their evolutionary history. However, the majority of their fossils are highly fragmentary, especially within sphenodontines—the group including its single surviving species, *Sphenodon punctatus* (the tuatara of New Zealand)—thus severely hampering our understanding on the origins of the tuatara. Here, we present a new sphenodontian species from the Early Jurassic of North America (Arizona, USA) represented by a nearly complete articulated skeleton and dozens of upper and lower jaws forming the most complete ontogenetic series in the sphenodontian fossil record. CT-scanning provides plentitude of data that unambiguously place this new taxon as one of the earliest evolving and oldest known sphenodontines. Comparisons with *Sphenodon* reveal that fundamental patterns of mandibular ontogeny and skeletal architecture in *Sphenodon* may have originated at least ~190Mya. In combination with recent findings, our results suggest strong morphological stability and an ancient origin of the modern tuatara morphotype.

[1] Museum of Comparative Zoology and Department of Organismic and Evolutionary Biology, Harvard University, Cambridge, MA 02138, USA. [2] Department of Earth and Environmental Sciences, Boston College, 140 Commonwealth Avenue, Chestnut Hill, MA 02467, USA. [3]Present address: School of Earth Sciences, University of Bristol, Queen's Road, Bristol BS8 1RJ, UK. ✉email: tsimoes@fas.harvard.edu; spierce@oeb.harvard.edu

Sphenodontians are one of the longest living lineages of extant reptiles, with a fossil record of at least 230 million years[1] and with recent morphological and molecular clock estimates suggesting their split from their closest relatives—squamates (lizards and snakes)—during the Late Permian at about 259 Mya[2]. Importantly, sphenodontians achieved a widespread geographic distribution between the Middle and early Late Triassic, with fossils recovered from various localities in the UK, USA, Germany, Brazil, Argentina, and Zimbabwe—e.g., refs. [1,3–8], indicating they quickly occupied northern and southern portions of Pangaea. Interestingly, sphenodontians sustained a higher taxonomic diversity compared to squamates during the Triassic and Jurassic, being surpassed by the latter as the most species-rich group of lepidosaur only in the Cretaceous[4]. This discrepancy in species richness between squamates and sphenodontians only increased during the Late Cretaceous and Cenozoic[9], culminating in the current 11,000+ species of extant squamates and only one sphenodontian species, *Sphenodon punctatus* (the tuatara of New Zealand)[10].

**Fig. 1 Holotype of *Navajosphenodon sani* (MNA.V.12442). a** As preserved in the sedimentary matrix in ventral view. **b** Micro CT-scanned and segmented whole skeleton in ventral view; **c** Micro CT-scanned and segmented whole skeleton in dorsal view (embedded within the sedimentary matrix). Ca.V. caudal vertebrae, Ce.V. cervical vertebrae, Do.V. dorsal vertebrae, Do.R. dorsal ribs, Fe femur, Fi fibula, H humerus, Ma manus, Pe.G. pectoral girdle, Pel.G. pelvic girdle, Ra radius, Ti tibia, Ul ulna. (l) left side and (r) right side. Note: this specimen was previously cataloged as MCZ VP 9016. Scale bar = 10 mm.

The higher taxonomic diversity of sphenodontians compared to squamates during the first half of their evolutionary history is contrasted by the quality of their fossil record. A considerable portion of sphenodontian fossil diversity is represented only by fragmentary specimens of little systematic value, such as isolated jaws and teeth—e.g., refs. [1,11–18]. This pattern has long posed a severe limitation to construct a robust hypothesis of sphenodontian phylogenetic relationships and evolutionary patterns—see discussions in refs. [3,19]—such as reconstructing the morphological changes leading to the considerable diversity of sphenodontian cranial morphotypes in the fossil record and the origin of the modern tuatara skull[20]. Most notably, there are almost no articulated skulls or postcrania in the fossil record of sphenodontines (but see ref. [21])—the branch of the sphenodontian tree including *Sphenodon*, and which is estimated to have originated during the Late Triassic at ~206 Mya[19]. During this 200+ Myr evolutionary history, all fossils robustly assigned to sphenodontines were comprised of jaw elements and isolated vertebrae, including all material referable to taxa such as *Cynosphenodon*, *Kawasphenodon*, *Sphenovipera*, among other unnamed Mesozoic specimens[12,14,15,22]. During the Cenozoic, this record became even poorer, with the entire sphenodontine fossil record being represented by a single lower jaw assigned to *Kawasphenodon* from Argentina[13], besides much younger Neogene fossils already identifiable as belonging to the extant *Sphenodon* in New Zealand[23]. Therefore, despite the longevity of sphenodontians and the relevance of the extant tuatara to broadscale evolutionary and conservation studies—e.g., refs. [2,24,25]—we have surprisingly little understanding of the evolutionary changes between early sphenodontians and *Sphenodon*.

Here we report on a new species of fossil sphenodontian represented by dozens of new specimens from the Early Jurassic of North America (Arizona, USA), including one individual represented by a nearly complete skeleton and which constitutes the most complete sphenodontian ever recovered from that continent. Using high-resolution micro-CT scanning and distinct phylogenetic methods, we find overwhelming support for the new species as one of the earliest evolving and oldest members of the tuatara lineage, revealing insights into the ontogeny and deep time evolutionary origins of the tuatara skull.

## Results

Systematic Palaeontology
Lepidosauria Dumeril and Bibron, 1839
Sphenodontia Williston, 1925 (*sensu* Simões et al., 2020)
Sphenodontinae Nopcsa, 1928
*Navajosphenodon sani* gen. et sp. nov.

**Etymology**. Genus epithet comes from a combination of "Navajo," in honor of the native people from North America that inhabit the Colorado Plateau where the specimens were found, and "sphenodon," in reference to the modern tuatara *Sphenodon punctatus*. Species epithet "sani," meaning "old age" in the Navajo language.

**Holotype**. MNA.V.12442 (previously cataloged as MCZ VP 9016), a fully articulated skeleton, including the skull, mandibles, axial and appendicular skeleton (Fig. 1).

**Referred materials**. MCZ VP 9098, MCZ VP 101562, MCZ VP 9099, MCZ VP 101564, MCZ VP 101575, MCZ VP 9094, MCZ VP 9102, MCZ VP 9103, MCZ VP 101569, MCZ VP 101563, MNA.V.8726(A-F), MNA.V.8727.

**Locality and horizon**. "Silty facies" of the Kayenta Formation, Glen Canyon Group—Gold Springs Quarry and Main Quarry, Adeii Eechii Cliffs, Coconino County, Arizona, USA. Sinemurian-Pliensbachian, Early Jurassic[26].

**Diagnosis**. Can be distinguished from all other species of sphenodontians by the following combination of features: premaxillae with no posterior maxillary process*; premaxillae with well-developed posterodorsal process; jugal with a wide posterodorsal process and a well-developed posteroventral process (contributing to a complete lower temporal bar); absence of a dorsolateral concavity on the surface of the postorbital; squamosal dorsal process bifurcated distally*; quadrate-quadratojugal completely fused; straight posterior margin of the quadrate-quadratojugal complex; presence of pterygoid teeth; presence of arcuate flanges on the pterygoids; dentary symphysial region small and slightly curved medially, premaxillary teeth present as discrete elements; quadrangular tooth bases for the additional tooth series; presence of caniniform successional teeth on the anterior end of the dentary; presence of posteromedially directed flanges on maxillary alternating teeth; presence of a midventral crest throughout the entire vertebral series; penultimate phalanges longer than preceding phalanges*; ungual phalanges tall at their bases. (*) Indicates features exclusively known in *N. sani* relative to all other sphenodontians.

**Morphological description and comparisons—skull**. The premaxillae are paired (Fig. 2), with the left premaxilla better preserved than its right counterpart and in articulation with the left maxilla (Fig. 3). The nasal process is slightly curved and directed posterodorsally (Fig. 3a), indicating the anterior end of the snout was smoothly curved and was not dorsoventrally deep as in clevosaurids[3,27,28]. The posterodorsal process of the premaxilla is extremely elongate, reaching posteriorly as far as the apex of the nasal (=facial) process of the maxilla and forming most of the posterior margin of the external nares. In contrast to most sphenodontians and other diapsid reptiles, the premaxillae do not possess a maxillary process distinct from the posterodorsal process extending posteriorly to contact the maxilla (Fig. 2). Instead, the maxilla reaches anteriorly to contact the main body of the premaxilla at the level of the distalmost premaxillary tooth. The left premaxilla has three acrodont teeth preserved in situ and not forming the ventrally expanded dentigerous beak observed in most sphenodontians—a condition more commonly observed in early sphenodontians, such as *Gephyrosaurus* and *Diphydontosaurus*[7,29] (and TRS, pers. obs.).

Both maxillae are preserved, with the right element split into two (Figs. 2 and 3b). The premaxillary process of the maxilla extends well anteriorly to contact the body of the premaxilla. The nasal process is relatively well-developed and descends anteriorly at a smooth angle towards the anterior extremity of the maxilla. Small mental foramina are observed along the edge of the dentigerous margin. At the base of the nasal process, a relatively larger foramen might represent the anterior superior alveolar foramen. The dentigerous portion of the maxilla extends well posteriorly on the skull to the level of the posterior margin of the orbit. The suborbital process of the maxilla thus forms most of the ventral margin of the orbit (more clearly visible on the right maxilla of MNA.V.12442). The holotype has nine teeth preserved on the right maxilla, with space allowing for at least five additional teeth, whereas the left maxilla has eight preserved teeth. Details on dental morphology, replacement, and ontogeny are further discussed below in the sections for "Dentition" and "Ontogeny of jaws and teeth."

Fragments of the paired nasals are preserved in MNA.V.12442 with the left element lying medially to the nasal process of the left premaxilla, whereas the right nasal bone is preserved medially to the broken apex of the right maxilla (Figs. 2 and 3d). Both are too poorly preserved to recognize how much they contribute to the medial margin of the external nares, but the available data suggests that the nasal process of the premaxilla was the main structure separating the external narial openings. A ventrolateral process of the nasal is present and preserved on the right element.

The prefrontals are dorsoventrally deep and form most of the anterior margin of the orbits (Figs. 2 and 3e), as in most sphenodontians, some stem lepidosauromorphs such as *Vellbergia*[30], and early-diverging squamate lineages (e.g., geckos). Dorsally, the prefrontal has an extensive articulation with the lateral margin of the frontal, reaching as far back as the midpoint of the orbit. Ventrally, the prefrontal contacts the ascending process of the palatine. There is no evidence of a lacrimal bone nor an articulatory facet on the lateral margin of the prefrontal for a lacrimal articulation and thus we consider the lacrimal to be absent, as in most other sphenodontians. The anteroventral margin of the prefrontal is slightly concave and forms a lateral opening for the exit of the lacrimal duct along with the dorsal margin of the maxilla, as observed in *Sphenodon*. The external surface has no visible sculpturing or a prefrontal crest, although the level of sculpturing may have changed in later ontogenetic stages.

One element preserved in between the nasals (dorsally) and the palatal region (ventrally) is interpreted here as the right septomaxilla (Figs. 2 and 3f). It is longer than wider and preserves a large facet that would have accommodated the vomeronasal organ.

The left postfrontal is located mostly ventral to the left frontal and separated from the postorbital, indicating it was displaced anteromedially (Fig. 2). The right postfrontal is located between the right postorbital and the right frontal. The postfrontal is tri-radiated with a relatively broad distal process that articulated with the anterior margin of the postorbital, thus providing a small contribution to the posterodorsal margin of the orbit (Figs. 2 and 3g).

The jugal is a robust element lying medially adjacent to the suborbital process of the maxilla and contributing to the ventral border of the orbit with the latter (Figs. 2 and 3h, j). The posterodorsal process of the jugal is considerably wide compared to most other diapsids, including other sphenodontians, and it bears along the orbital margin an articulatory facet for articulation with the ventral process of the postorbital (Figs. 2 and 3k). The posteroventral process is extremely well-developed and the right jugal on MNA.V.12442 suggests this process extends posteriorly to the level of the quadrate-quadratojugal, thus forming a complete (or nearly complete) lower temporal bar (Figs. 2 and 3h). As the degree of development of the posteroventral process of the jugal is known to be quite variable throughout postembryonic ontogeny both in early sphenodontians[7,27] and squamates with a lower temporal bar[31]— besides considering the subadult stage of the holotype (see "Ontogeny of jaw and teeth")—it is expected that most subadult and adults would have had a complete lower temporal bar, although the condition remains unknown in young juveniles and hatchlings.

The quadrates and quadratojugals form a single fused quadrate-quadratojugal complex with no suture being visible between both elements (Figs. 2 and 3i), as similarly observed in later ontogenetic stages of the modern *Sphenodon*. A quadratojugal fenestra is seemingly present between the quadrate and quadratojugal and the posterior margin of the quadrate is pillar-like and relatively straight, as observed in *Sphenodon*, instead of forming a posterior emargination as in early evolving sphenodontians—e.g., *Gephyrosaurus* and *Diphydontosaurus* (TRS, pers. obs. and refs. [7,29]). Medially, the quadrate-quadratojugals have a

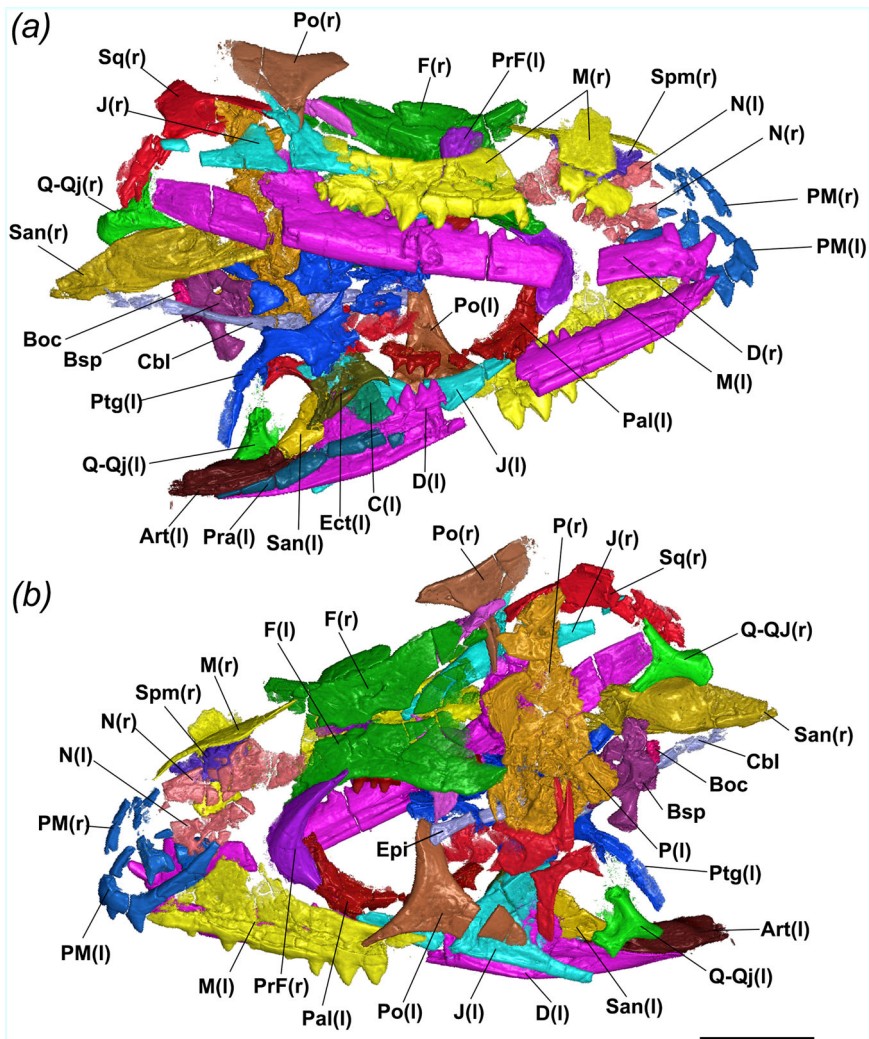

**Fig. 2 Micro CT-scanned and fully segmented skull and mandibles of the holotype of *N. sani* (MNA.V.12442). a** Right ventrolateral view. **b** Left dorsolateral view. Art articular, Boc basioccipital, Bsp basisphenoid, C coronoid, Cbl first ceratobranchial, D dentary, Ect ectopterygoid, Epi epipterygoid, F frontal, J jugal, M maxilla, N nasal, P parietal, Pal palatine, PFr postfrontal, PM premaxilla, Po postorbital, Pra prearticular, PrF prefrontal, Ptg pterygoid, Q-Qj quadrate-quadratojugal, San surangular, Spm septomaxilla, Sq squamosal, (l) left side and (r) right side. Scale bar = 1 mm.

well-developed anteromedially directed and dorsoventrally deep process to contact the pterygoids. Two condyles are distinct on the ventral margin of the quadrate-quadratojugal and are in close proximity to the glenoid facet of the articular bone in the lower jaw.

The postorbitals are preserved on both sides of the skull in MNA.V.12442, with the left postorbital nearly completely preserved (apart from a crack dividing its posterior process) and in close contact with the jugal, maxilla, squamosal, and postfrontal (Fig. 2). The postorbitals are triradiate, with slender dorsal and ventral processes, and a much more dorsoventrally deep and elongate posterior process for the contact with the squamosal (Figs. 2 and 3k). The anteromedial margin of the dorsal and ventral processes of the postorbital preserved the articulatory facets for postfrontal and jugal, respectively.

The squamosals are preserved on either side of the skull in MNA.V.12442 with the left element in a much better state of preservation (Fig. 2). The squamosals are tetraradiate as in other sphenodontians and many other early evolving diapsid reptiles—e.g., *Youngina* and *Prolacerta* (TRS pers. obs.), and differing from the triradiate condition observed in early evolving squamates, such as *Megachirella* and *Huehuecuetzpalli*[32,33]. The squamosal has a strongly developed dorsal process, which would have

contacted the supratemporal process of the parietal, and which is bifurcated distally (Figs. 2 and 3l). The anterior process is dorsoventrally deep, elongate, and bears a deep lateral parabolic-shaped articulatory facet for the reception of the posterior process of the postorbital. The anteroventral process is very elongate, being much longer than the posteroventral process, with both processes forming a dorsal cap that is likely to have embraced the quadrate-quadratojugal element as in the modern *Sphenodon*.

The frontals are paired and are well-preserved along the orbital margins, but their anterior and posterior margins are shattered (Figs. 2 and 3m, n). The frontals have greatly elevated anterolateral articulatory facets to receive the prefrontal dorsal process and elongate posteroventrolateral articulatory facets to receive the postfrontal (Figs. 2 and 3o). Both suggest that the frontals contributed little to the dorsal margin of the orbits. On their ventral side, the frontals have weakly developed subolfactory processes. A small level of sculpturing is observed on the external surface of the frontals.

The parietals are both poorly preserved and in tight articulation with each other (Figs. 2 and 3m). Although the anterior margin of the left parietal is preserved, it was bent dorsally and thus the nature of its contact with the left frontal cannot be determined. The left element suggests the parietals are

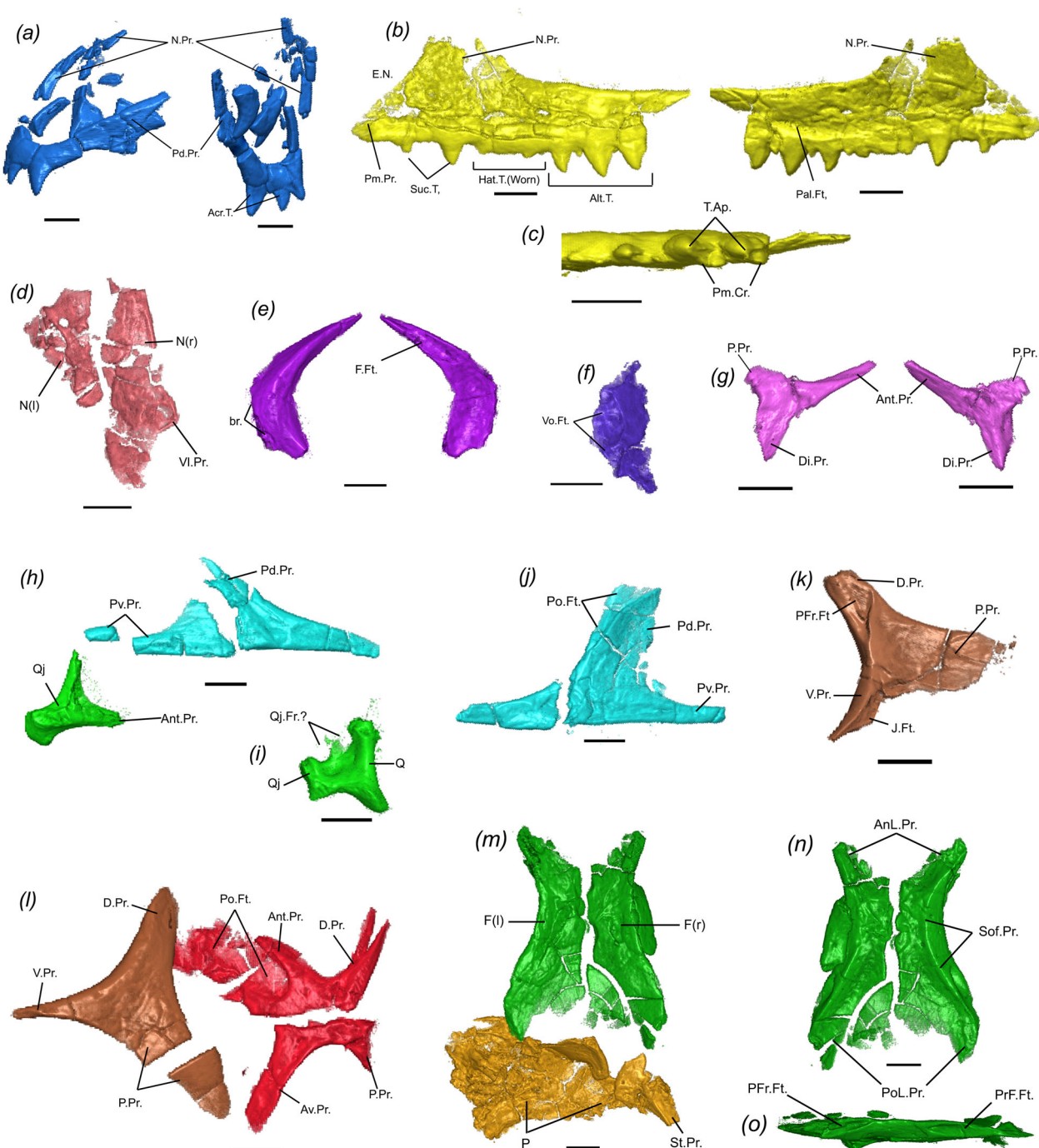

**Fig. 3 Individual elements of the skull of the holotype of *N. sani* (MNA.V.12442). a** Premaxilla in lateral (left) and posterior (right) views. **b** Left maxilla in lateral (left) and medial (right) views. **c** Left maxilla in occlusal view. **d** Nasals in dorsal view. **e** Left prefrontal in lateral (left) and medial (right) views. **f** Right septomaxilla in ventral view. **g** Right postfrontal in lateral (left) and medial (right) views. **h** Right jugal and associated quadratojugal in lateral view. **i** Fused left quadrate–qudratojugal complex in posterior view. **j** Left jugal in lateral view. **k** Right postorbital in medial view. **l** Left postorbital and squamosal in lateral view. **m** Frontals and parietals in dorsal view. **n** Frontals in ventral view. **o** Right frontal in lateral view. Relative bone positions in **h**, **l**, **m** are as preserved in the specimen. Ant.Pr. anterior process, Av.Pr. anteroventral process, Acr.T. acrodont teeth, Ant.Pr. anterior process, AnL.Pr. anterolateral process, br. broken edge, D.Pr. dorsal process, Di.Pr. distal process, E.N. external nares, J.Ft. facet for jugal, N.Pr. nasal process, P.Pr. posterior process, Pal.Ft. facet for maxillary process of palatine, PFr.Ft. facet for postfrontal, Pm.Cr. posteromedial crest, Pm.Pr. premaxillary process, Po.Ft. facet for postorbital, Po.Pr. postorbital process, PoL.Pr. posterolateral process, PrF.Ft. facet for prefrontal, Pv.Pr. posteroventral process, Q quadrate, Qj quadratojugal, Qj.Fr. quadratojugal foramen, Sof.Pr. subolfactory process, St.Pr. supratemporal process, T.Ap. tooth apex, V.Pr. ventral process, Vl.Pr. ventrolateral process, Vo.Ft. facet for vomeronasal organ. Scale bars = 1 mm.

broad, resembling the condition on *Homeosaurus* (TRS, pers. obs. and[34]), but this condition may be the result of compression of the specimen. There is no evidence for a developed sagittal crest, suggesting this feature was either absent or weakly developed in MNA.V.12442. Considering the parietals experience a great degree of ontogenetic change in extant and fossil lepidosaurs[31,35,36], we consider that additional specimens with undistorted parietals and from a later ontogenetic stage would be necessary to fully understand the morphology of this element. The posterior margin of the parietals is poorly preserved and the contact with the supraoccipital cannot be determined.

Both palatines are preserved on the ventral side of the skull (Fig. 4), mostly represented by the well-developed single palatine tooth row with teeth of similar size to the anterior teeth in the maxillae and dentaries (Figs. 2 and 4a, b). The palatine has a well-developed dorsal process that contacts the prefrontal dorsally, and the maxilla and jugal laterally. Owing to poor preservation, it is not possible to determine if the palatines met at the midline anteriorly, as in most sphenodontians.

The pterygoids are mostly preserved on their posterior portion, including the transverse and the quadrate processes (Figs. 2 and 4a, b). The transverse process is broad and contacts the ectopterygoid distally, but it is not possible to tell the degree of development of the transverse pterygoid flanges. Both pterygoids have well-developed arcuate flanges on the midline for the articulation with the basipterygoid processes of the basisphenoid. The quadrate processes are elongate and curved posterolaterally, being located medially to the pterygoid process of the quadrate-quadratojugal. The body of the pterygoid features small denticles organized in at least one dental row.

The ectopterygoid is better preserved on the left side of the skull (Figs. 2a and 4a, b). The element is mostly distorted but seems to have had a broad articulation with the transverse process of the pterygoid and a well-developed posterolateral process.

The left epipterygoid is preserved ventral to the left parietal and exiting the skull on the space between the left parietal and left postorbital (Figs. 2 and 4a). It is delicate and rodlike with an expanded base as in most sphenodontians and early diapsids. The first ceratobranchial is preserved on the right side of the skull extending from the main body of the pterygoids to the posterior end of the right surangular (Fig. 2).

Very few of the braincase elements are preserved or diagnosable. The most informative component is the basisphenoid, located immediately posterior to the pterygoids and ventrally to the parietals' posterior margin (Fig. 2). It has well-developed basipterygoid processes with distally expanded elliptical articulatory surfaces for the pterygoids (Fig. 4c–f). In ventral view, a short cultriform process extends anteriorly and includes a pair of short trabeculae cranii dorsally. A pair of ventral openings represent the entrance of the internal carotids. In dorsal view, the dorsum sellae is exposed forming the posterior margin of a deep pituitary fossa lying on the well-developed sella turcica. Anteriorly, the abducens canals for the cranial nerve VI are located dorsolaterally to the cultriform process and ventrally to the clinoid processes. Overall, these features are similar to the braincase anatomy of many other lepidosaurs, including terrestrial fully-limbed squamates, with the exception of the ventral location of the openings for the carotids[37,38]. A small portion of the basioccipital is preserved in tight articulation with the posterior margin of the basisphenoid.

**Mandibles**. Smaller specimens (inferred as juveniles, see below) have a relatively lower degree of ossification and are slender relative to the dentaries of MNA.V.12442 (Fig. 5), which in turn, is comparatively slender in relation to the largest specimens (inferred as adults) (see "Ontogeny of jaws and teeth", below). As in other sphenodontians, the dentaries have a strongly developed and ascending coronoid process, as well as a very elongate posteroventral process extending far posteriorly, to the level of the glenoid in MNA.V.12442 (Fig. 5a). The Meckelian canal is open medially and is relatively shallow. The ventral margin of the dentaries curve inwards and upwards creating a ventral crest that forms the ventral border of the Meckelian canal (Fig. 5b, c). The dentary symphyses are not clearly visible in most specimens, but in the holotype, they are small, nearly straight, and slightly expanded dorsoventrally (Fig. 5c), thus differing from the strongly curved condition in *Gephyrosaurus* and *Diphydontosaurus* (TRS, pers. obs. and refs. [7,29]). It is possible that the more robustly built mandibles in the larger specimens would bear a differently shaped symphysis, especially a greater degree of dorsoventral elongation as implied by their comparatively deeper dentary. For details on dental morphology see section "Dentition" below.

The surangular on the right mandible was displaced ventrally relative to the posteroventral process of the dentary, revealing a deep articulatory face for the latter on its lateral surface and indicating that the surangular extended at least to the level of the coronoid process anteriorly (Fig. 2). The left surangular is preserved in articulation with the other postdentary bones, with only a small portion of it being exposed in medial view. It forms the dorsal margin of the mandibular adductor fossa, and its anterior end ascends dorsally to contribute to the coronoid process (Figs. 2 and 5d).

The coronoid is reduced as in most sphenodontians, contributing only to the anteromedial portion of the coronoid process (the rest composed by the dentary and surangular) (Figs. 2 and 5d). The coronoid dorsal process is short and, when in articulation, would not have ascended above the level of the coronoid process of the dentary, thus being mostly hidden in lateral view. An anteromedial and a smaller posterodorsal process are present on the left coronoid, thus contrasting with the condition in *Sphenodon* in which those are highly reduced or absent. The prearticular is elongate, contributing to the retroarticular process ventrally and extending anteriorly at least to the level of the coronoid bone—a small anterior fragment likely belonging to the prearticular indicates it could have reached the level of the posteriormost dentary tooth (Figs. 2 and 5d). The prearticular forms the ventral margin of the mandibular adductor fossa and contacts the other postdentary elements ventrally, except for the coronoid. The angular is not preserved in the holotype, but it is slightly exposed in lateral view on MCZ VP 101569 (Fig. 6). The articular is not fused to any of the other postdentary bones, forming most of the glenoid surface and retroarticular process. It does not have any discernible processes projecting medially or laterally. The dorsal surface of the glenoid has an elongate central ridge for articulation with the quadrate, which would have enabled propalinal movement of the lower jaw (Fig. 5d).

**Dentition**. Different dental types and their changes throughout ontogeny are discussed below in the section "Ontogeny of jaw and teeth" (Fig. 6). Here we provide additional information pertaining to the individual morphology of distinctive dental types.

The premaxillary teeth are only observed in the holotype (Figs. 2 and 3a), and at this stage, the teeth are fully "acrodont": they are placed apically and fully ankylosed to the jawbone. Three individual teeth of distinct sizes are preserved on the left premaxilla instead of forming a single chisel-like premaxillary tooth, similar to early forms such as *Diphydontosaurus*, *Planocephalosaurus*, *Clevosaurus hudsoni* and also *Homeosaurus* (TRS,

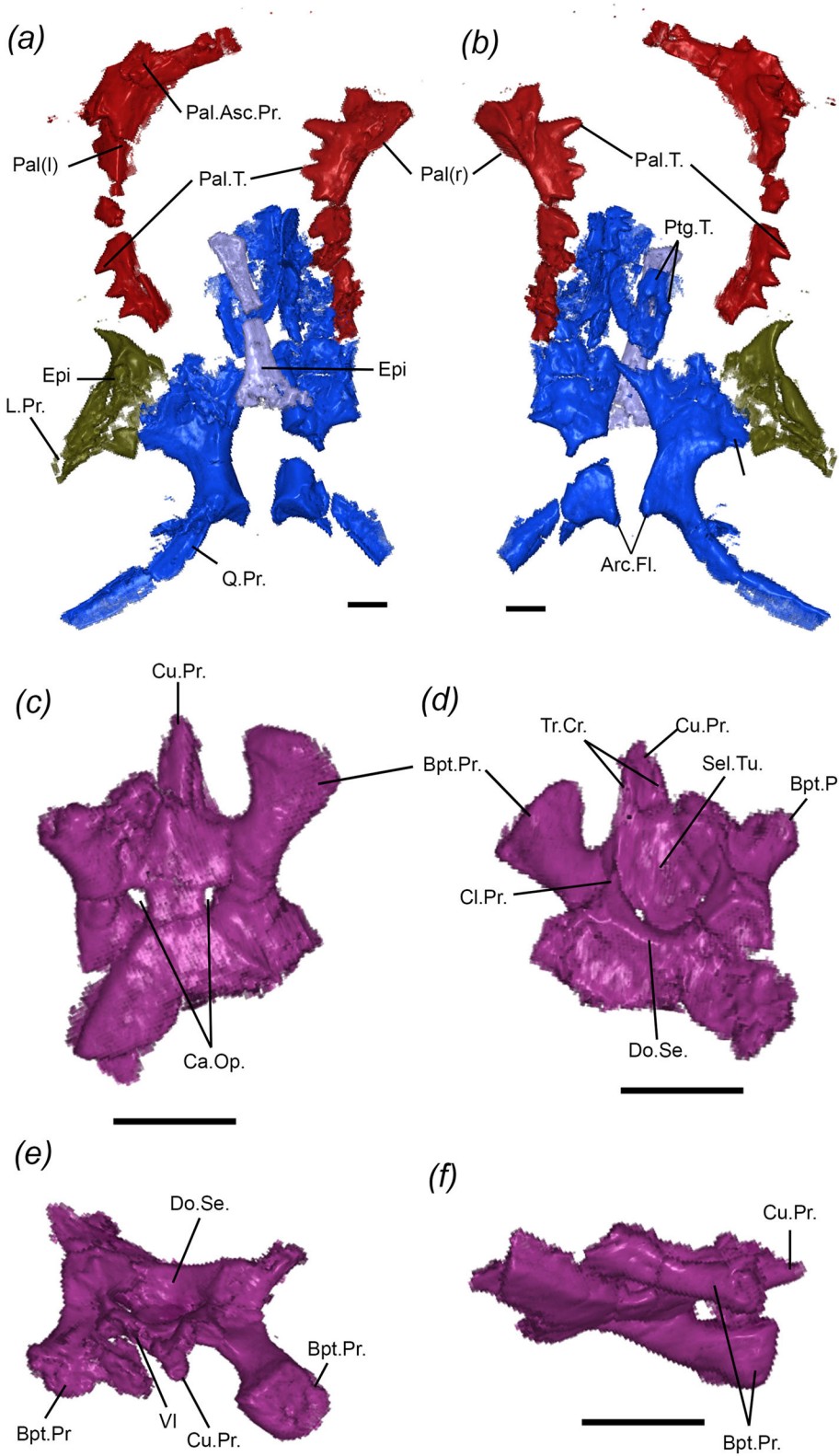

**Fig. 4 Palatal region and braincase of the holotype of *N. sani* (MNA.V.12442).** Palatal region in dorsal (**a**) and ventral (**b**) views. Basisphenoid in ventral (**c**), dorsal (**d**), anterior (**e**), and right lateral (**f**) views. Arc.Fl. Arcuate flanges, Bpt.Pr. basipterygoid process, Ca.Op. openings for internal carotid arteries, Cl. Pr. clinoid process, Cu.Pr. cultriform process, Do.Se. dorsal sella, L.Pr. lateral process, Pal.Asc.Pr. palatine ascending process, Pal.T. palatine teeth, Ptg.T. pterygoid teeth, Q.Pr. quadrate process, Sel.Tu. sella turcica, Tr.Cr. trabeculae cranii, VI passage for cranial nerve VI (abducens canal). Scale bars = 1 mm.

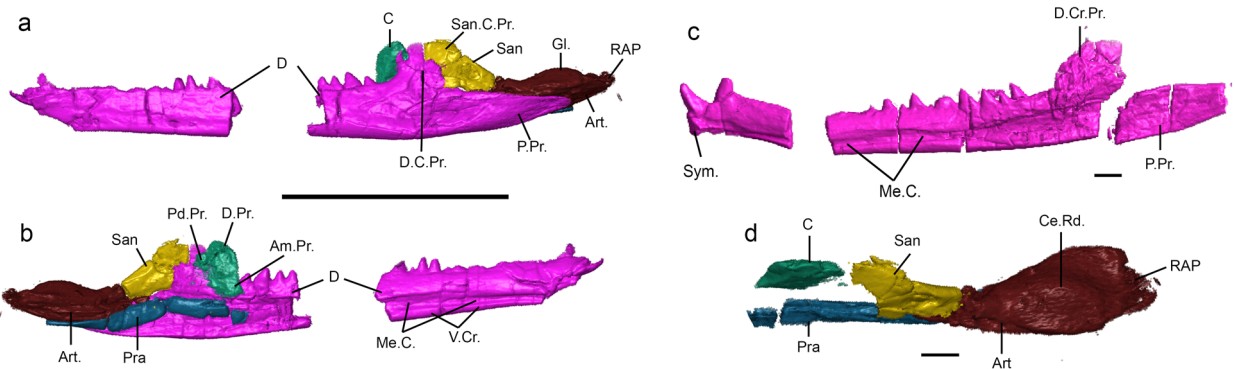

**Fig. 5 Mandibles of the holotype of *N. sani* (MNA.V.12442).** Left mandible in lateral view (**a**) and medial view (**b**). Right dentary in medial view (**c**). Left post-dentary bones in dorsal view (**d**). Ce.Rd. central ridge, Gl. glenoid articulation, Me.C. Mekelian canal, RAP retroarticular process, San.C.Pr. surangular coronoid process, Sym. symphysis, V.Cr. ventral crest. Scale bars = 10 mm (**a**, **b**) and 1 mm (**c**, **d**).

pers. obs and refs. [7,27,34,39]). No accessory flanges are observed in any premaxillary tooth.

Nearly all identified maxillae have quite distinct dental regions as typically found in sphenodontines, including large anteriorly located successional teeth, followed distally by a region of hatchling teeth that is heavily worn in most individuals, which in turn is followed distally by an alternating tooth series. The alternating teeth are robust, slightly inclined posteriorly, bearing a single enlarged cusp. Further, alternating teeth include a posteromedially directed dental crest (= dental flange), as also observed in sphenodontines. The additional tooth series is represented by three to five smaller teeth located distally to the alternating tooth series in the smallest individuals (Fig. 6g). In the largest individuals (Fig. 6j), most additional teeth comprise the largest teeth on the tooth row, with the posteriormost two additional teeth being smaller than the anterior ones. The presence of a posteromedially directed flange could not be confirmed on the additional teeth.

Dentary teeth include a pair of anteriorly located canine-like successional teeth that represent the largest teeth in the tooth row among the largest individuals (Figs. 2, 3, and 6). Distally to the successional teeth, hatchling and alternating teeth occur in the smaller specimens (Fig. 6j, k). The latter two dental series are gradually worn away with increasing dentary size (interpreted as an ontogenetic feature, see below), eventually forming an edentulous region where teeth have been completely worn-away, as also observed in the hatchling tooth series and part of the alternating series of *Sphenodon* (Fig. 6e, f) and *Cynosphenodon*[14]. The alternating teeth are similar in shape to the maxilla alternating teeth, with the larger teeth being robust and conical, but having their distal margin longer than the medial margin, making the teeth look inclined anteriorly when observed in lateral or medial view (Fig. 6k). The additional tooth series makes up most of the dentary tooth row in adult individuals (Fig. 6l), and similarly to the alternating teeth, appear to be inclined anteriorly due to the more convex distal tooth margin. Further, additional teeth bear an incipient posterior flange, which becomes reduced (possibly due to tooth wear) in older adults (Fig. 6e, f). This flange is much smaller than that observed in young adults of *Sphenodon* and other sphenodontines, in which they also become harder to detect in older individuals owing to dental wearing (TRS pers. obs.; FMNH 11113, MCZ VP R4702).

**Postcranium**. The holotype reserves most of the postcranium in full articulation, including vertebrae, ribs, pectoral, and pelvic girdles, as well as fore and hind limbs. All other referred specimens either do not possess postcranial elements, or they are poorly preserved. The postcranium description below is, therefore, entirely based on the holotype (Fig. 7).

**Axial skeleton**. There are 21 presacral vertebrae preserved, including 4 cervicals and 17 dorsals, but the total number probably reached closer to 25 in MNA.V.12442 (including at least two additional cervicals). At least one sacral is visible ventral to the pelvic girdle in the CT scans and the pelvic region is followed by eight caudals preserved in partial articulation. The posteriormost caudal region is not preserved in MNA.V.12442.

The atlas and axis are preserved in the holotype (Fig. 7a, b), including an atlas centrum (odontoid) that is not fused to the axial pleurocentrum, which further suggests the juvenile stage of this specimen. The atlas neural arches are preserved dorsal to the axial pleurocentrum, with one of the neural arches broken into two separate components. The axis neural arches were not fused to the centrum body (another juvenile feature). The axis and the third and fourth cervicals have a distinct midventral crest on the ventral side of the centrum, with a pair of nutrient foramina on each side.

Both cervical and dorsal vertebrae are very similar in morphology. The pleurocentra are amphicoelic with deep cotyles and an open notochordal canal, as in most sphenodontians. The cervical and anterior dorsal pleurocentra have elliptical shapes in cross-section at the level of the cotyles, being even narrower on the midpoint of the vertebrae even when discounting some degree of taphonomic deformation. The posterior dorsals have more circular shapes in cross-section and so do the caudal pleurocentra. The ventrolateral sides of the pleurocentra are slightly concave and they meet ventrally forming a midventral crest along the entire vertebral series throughout the dorsal series (Fig. 7c).

The prezygapophysis and postzygapophysis are well developed, but there are no visible signs of accessory neural arch articulations (zygosphenes-zygantra)—but we note that their presence cannot be entirely excluded. The neural canal is relatively small in diameter and the neural arches at the level of the cervicals have approximately half the height of the pleurocentra. Among the dorsals, the neural arches increase in height, being at least as tall as the pleurocentra. The neural spine is relatively short in the cervical region, becoming much taller in the dorsal region. The diapophyses are connected to the parapophyses by a small ridge, forming an elongate and obliquely oriented synapophysis, as in most other sphenodontians. The intercentra are observed in the cervical region and located in the intervertebral space.

Caudal vertebrae include mostly the anteriormost region, represented by pygals. Some of the posteriormost pygals include an autotomy septum (Fig. 7d, e), which is characteristic of most sphenodontians and squamates.

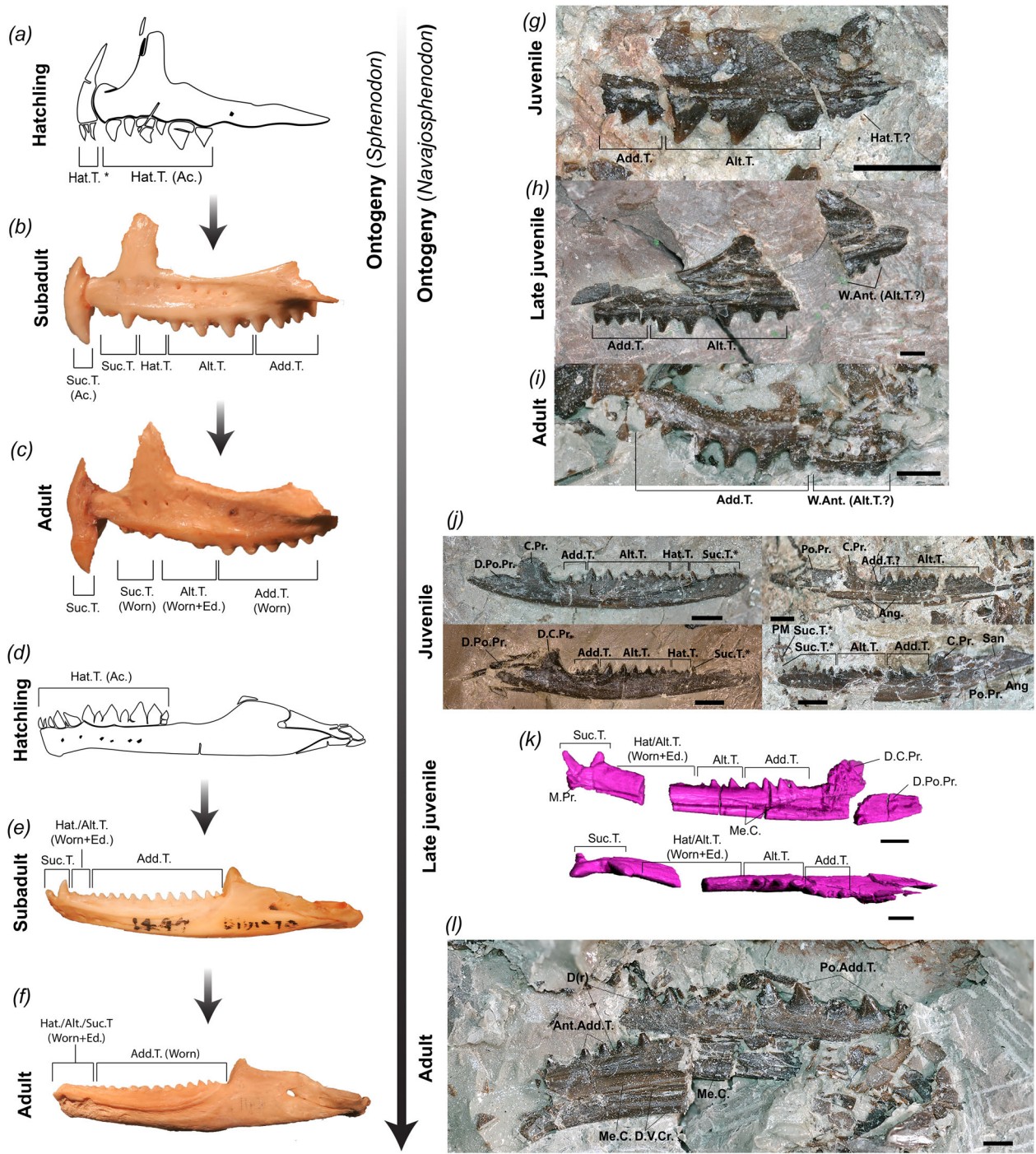

**Fig. 6 Conservation of ontogenetic stages from *Navajosphenodon* to *Sphenodon*.** Ontogenetic series on the maxilla (**a–c**) and dentary (**d–f**) of Sphenodon punctatus (rescaled to the same length). Ontogenetic series on the maxilla (**g–i**) and dentary (**j–l**) of Navajosphenodon sani. Specimen numbers: FMNH 207433 (**b**, **e**); FMNH 11113 (**c**, **f**); MCZ VP VP 9094 (**g**); MCZ VP VP 9100 (**h**); MCZ VP VP 9093 (**i**); MCZ VP VP 101564 (top left), MCZ VP VP 9094 (top right), MCZ VP VP 9099 (bottom left), MCZ VP VP 101569 (bottom right) (**j**); MNA.V.12442 (**k**); MCZ VP VP 9093 (**l**). **a**, **d** Re-drawn from ref. [43]. Add.T. additional teeth, Alt.T. alternating teeth, Ang angular, D.C.Pr. dentary coronoid process, D.Po.Pr. dentary posterior process, Ed. edentulous region, Hat.T. Hatchling teeth, M.Pr. mentonian process, Me.C. Meckelian canal, Suc.T. Successional teeth, Worn worn out teeth. Scale bars = 1 mm.

The cervical and dorsal ribs are elongate, narrow, and occur throughout the entire presacral region with no indication of a lumbar region. Rib heads are wide with poorly differentiated tuberculi and capitula. The anteriormost caudal ribs (pygal region) are fused to the pleurocentra being articulated to synapophyses on the remaining of the caudals.

There are no traces of a mineralized presternum, but we cannot rule out its presence on the holotype.

**Appendicular skeleton**. The pectoral girdle is poorly preserved, but remains of both coracoids and scapulae are present (Fig. 7h). The scapula is relatively tall, narrow, and straight. The anterior surface of both scapulae is poorly preserved and there are no signs of an anterior emargination. The preserved portion of the coracoids are smaller than the scapulae, do not possess any anterior emarginations, and the supracoracoid foramen is observable in at least one of the preserved elements. The glenoid is well-defined

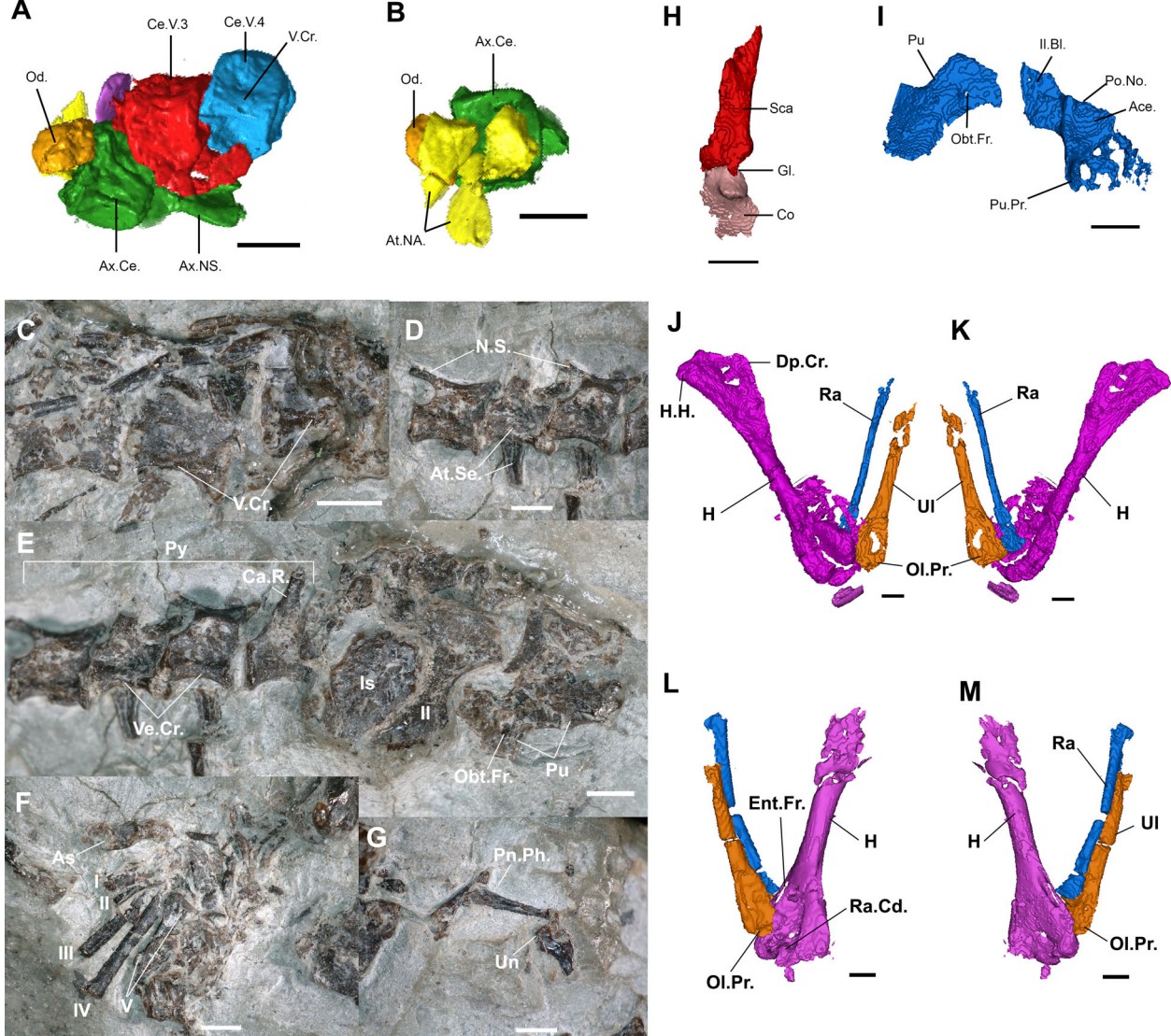

**Fig. 7 Postcranial skeleton of the holotype of *N. sani* (MNA.V.12442). a** Atlas, axis, and cervical vertebrae 3 and 4 in ventral view. **b** Part of Atlas–axis complex in dorsal view. **c** Dorsal vertebrae in ventral view. **d** Caudal vertebrae in lateral view. **e** Pelvic girdle and pygals in ventrolateral view. **f** Right pes in partial articulation. **g** Left pes with only distal phalanges preserved. **h** Left scapula and coracoid in lateral view. **i** Right(?) pubis and ilium. **j** Left forearm in posterior view. **k** Left forearm in anterior view. **l** Right forearm in anterior view (slightly displaced radius and ulna). **m** Right forearm in posterior view. Ace acetabulum, As astragulus, Ax axis, Ax.Ce. axis pleurocentrum, Ax.NS. axis neural spine, Ca.V. caudal vertebra, Co coracoid, Ce.V. cervical vertebra, Dp.Cr. deltopectoral crest, Ent.Fr. entepicondylar foramen, H humerus, H.H. humeral head, Il ilium, Il.Bl. iliac blade (partially preserved), Is isquium, N.S. neural spine, Obt.Fr. obturator foramen, Od. odontoid (atlas centrum), Ol.Pr. olecranon process, Pn.Ph. penultimate phalanx, Po.No. posterior notch on acetabulum, Pu pubis, Pu.Pr. anterior pubic process, Py pygals, Ra radius, Ra.Cd. radial condyle, Sca scapula, Ul ulna, V.Cr. midventral crest, I–V digit number. Scale bars = 1 mm.

and formed with equal contributions from the scapula and coracoid. There are no traces of an epicoracoid mineralized cartilage associated with the ossifications. No remains of the clavicles and interclavicle were detected. In the pelvic girdle, the left ischium, ilium, and pubis are preserved in close proximity (Fig. 7e), and parts of the right pubis and ilium embedded in the matrix were visible through CT scanning (Fig. 7i). The pubis is characterized by an obturator foramen close to the acetabulum and a ventrodistally expanding medial margin. The ilium is narrow and elongate with a relatively small iliac blade. The ilium is comprised of a large bony plate, but its medial margin is broken and the presence of an ischiadic tuberosity cannot be assessed.

Both forelimbs are preserved in close association with the pectoral girdle and to each other (Figs. 1 and 7j–m). The humeri are straight, with their distal ends slightly twisted relative to the proximal ends (at approximately 30°). The proximal ends are poorly preserved, and the shape of the humeral head is not clear, but the deltopectoral crest is moderately developed. The distal end of the humerus is expanded and larger relative to the humeral head, bearing an entepicondylar foramen that is completely open (connecting the ventral and dorsal sides of the humerus) (Fig. 7l, m). The ectepicondylar foramen is not clearly visible. A moderately developed capitulum for connection with the radius is observed on the right humerus (Fig. 7l, m).

The radii are slender and elongate, located just next to the preserved ulnae. Their proximal ends have a shallow concavity for articulation with the humeral capitulum. The ulnae have a moderately developed and ossified olecranon process proximally, but their distal ends are not preserved on either side (Fig. 7j–m). The ulnae are elongate and stouter than the radii.

The left pes is partially preserved in close proximity to the distal ends of the left tibia and fibula with the proximal part of the astragalus exposed, along with the five metatarsals (Fig. 7f). The fifth metatarsal is partially obscured by matrix and its head is expanded, but with only a moderate observable degree of inflection, thus constituting a weakly-hooked fifth metatarsal. The right pes is poorly preserved, except by one digit (Fig. 7g), which includes an elongate penultimate phalanx and an ungual phalanx relatively tall (dorsoventrally deep) at its base.

**Ontogeny of jaws and teeth.** In hatchlings of *Sphenodon* the dentition is already placed apically and ankylosed to the jaw bones (i.e., acrodont)[40–42]—although the premaxillary dentition may not yet be entirely ankylosed[43] (Fig. 6a, d). In older juveniles and adults, all teeth are externally visible as apically placed and ankylosed to the jaw bones, but it has been previously noted that they appear to be located in shallow sockets in adults—a condition previously termed hyperacrodonty[40]. Indeed, it was subsequently found in the cross-section of fossil sphenodontians that teeth can be deeply rooted into the jaws, such as in *Cynosphenodon* and *Priosphenodon*[44,45].

In terms of replacement patterns, there are five generations of teeth in *Sphenodon*, the first three occurring in the embryo and representing small alternating teeth. The fourth and fifth generations occur after hatching and represent the anteriormost teeth on the mandibles, maxillae, and premaxillae—termed successional (or replacement) teeth[7,40–42,46] (Fig. 6b, e). They are larger and so they replace two or more teeth from the previous generation[40]. Those may fuse to each other in each premaxilla forming a chisel-shaped structure. Some of the hatchling dentition is not replaced in the maxilla, forming a short series of teeth between the larger anterior successional teeth and the subsequent alternating tooth series—termed hatchling teeth. Further posteriorly on the maxilla, small teeth from older generations are kept with larger successional teeth[41]. These different generations of teeth produce a pattern of alternating teeth[7,40–42] (Fig. 6b, e). Finally, teeth homogeneous in size are continuously added posteriorly to the lower and upper jaws as they grow and are termed additional teeth (sometimes referred to as uniform teeth)[7,40–42,46]. This last process begins much earlier in the dentary relative to the maxilla, and consequently, there are more additional teeth on the dentary of adult forms than in the maxilla[40] (Fig. 6a–f). In embryos, however, the alternating pattern is clearly visible in both upper and lower jaws (Fig. 6a, d). Teeth on the mandible grow in proportion to the increase in the length of the jaw[47], so the dentary teeth are uniformly increasing in size posteriorly (Fig. 6c, f).

The mandibular elements from *N. sani* reported here represent individuals that come from the same locality but are quite variable in size, with the largest individuals having dentary lengths nearly twice the size of the smallest individuals (Supplementary Table 1). Along with additional ontogenetic markers (Fig. 6), these specimens are best interpreted as comprising an ontogenetic series. Specifically, the smallest and best-preserved individuals (inferred juveniles) are relatively similar in size and include all dental categories normally recognized in hatchlings and juveniles of *Sphenodon*: successional, hatchling, alternating, and additional teeth series. Among those, there are several small-sized successional teeth preserved in situ—at least six in MCZ VP 101569—and space for a similar number of successional teeth in other small individuals—i.e., MCZ VP 9099, 9094, and 101564 (Fig. 6g, h, j). The two best-preserved juveniles (MCZ VP 9099 and 101564) have a short series of hatchling teeth with three or four teeth found in situ (Fig. 6j). The alternating tooth series comprises a series of five to

seven larger teeth interspaced by smaller teeth. Finally, the additional series in those individuals is comprised of only two to four additional teeth.

The holotype (MNA.V.12442) is an intermediately sized individual that is ~30% longer than the juvenile dentaries, but less than 50% of the length of the largest dentary (Supplementary Table 1 and Fig. 6j–l). It has at least five additional teeth, it already demonstrates some degree of tooth wear on the dentary anteriorly, and it bears only two successional teeth on the dentaries (Fig. 6k). As in *Sphenodon*, the maxillae on the holotype retain many teeth from younger ontogenetic stages, thus being composed of a couple of anterior successional teeth, a small section of hatchling teeth, and a posteriorly dominant alternating tooth series (Fig. 3b, c). Additionally, this specimen has the odontoid still unfused to the centrum body of the axis, further suggestive of skeletal immaturity (see more below).

The largest individuals (dentaries about twice as long as the juvenile dentaries) have dentaries that are more robustly built and dorsoventrally deep than smaller specimens (Fig. 6l). They include a much longer series of additional teeth that form most of the tooth row on both dentaries and maxillae (Fig. 6i, l). The anteriormost region of the adult dentaries is not preserved in any specimen, but the anteriormost preserved teeth are much smaller in size compared to the last additional teeth on the tooth row, further suggesting a later ontogenetic stage of these specimens. The anteriormost preserved region of the dentary in MCZ VP 9093 indicates a great degree of tooth wear (being nearly edentulous), and impressions on the sedimentary matrix indicate the presence of two large anterior successional teeth (Fig. 6l).

This pattern of size, bone shape variation, and changes in dental categories observed among all sampled *N. sani* individuals thus closely matches the ontogenetic sequence of dentary and maxillary changes in *Sphenodon*. Most of the isolated jaw elements recovered are, therefore, interpreted as belonging to young juveniles (although no hatchlings seem to be present), some older juveniles intermediate in length between the youngest individuals and adults (e.g., MNA.V.12442), and adult individuals with dental features and dental wearing typically found in older adults of *Sphenodon* (Fig. 6).

**Comparative anatomy and taxonomy.** *N. sani* shares with sphenodontids the straight posterior margin of the quadrate-quadratojugal complex and the presence of arcuate flanges on the pterygoids. Additionally, *N. sani* shares with sphenodontines the caniniform successional teeth anteriorly on the dentary, the quadrangular shape of the additional teeth bases, and the anterior process of the quadratojugal (further suggestive of a complete lower temporal bar). Further, the worn-out dentition at the position of the hatchling and alternating teeth series on the anterior portion of the maxillae and dentaries among larger sized forms (i.e., adults as defined here) is similar to patterns of dental regionalization and tooth wear observed in *Sphenodon* (Figs. 6 and 7). However, *N. sani* differs from most sphenodontines by retaining the pterygoid dentition, the premaxillary dentition present as discrete elements (instead of a single chisel-shaped tooth on each premaxilla), a posterodorsally elongate process of the premaxilla, and the absence of a dorsolateral concavity on the surface of the postorbital—all more commonly observed among earlier evolving, non-sphenodontid sphenodontians. Finally, the presence of a bifurcated dorsal process of the squamosal is unique to *N. sani* among all sphenodontians currently known. The combination of these features suggests sphenodontine affinities of *Navajosphenodon*, but as an early evolving form still retaining plesiomorphic sphenodontian features (confirmed by the phylogenetic analyses— below).

Among other Mesozoic sphenodontines, *N. sani* differs from *Cynosphenodon*[14,44] by having two caniniform successional teeth, a much reduced mentonian process, and the Meckelian canal curving anteroventrally on the dentary, instead of nearly closed by an expanded ventral crest of the dentary. Compared to *Kawasphenodon peligrensis*[13] and *K. expectatus*[22], *N. sani* has a more smoothly inclined coronoid process of the dentary, an elongate and relatively straight dental margin of the dentary instead of a short and concave dental margin, a much higher tooth count, and a distinct orientation of the dentary teeth (mesiodistally oriented with short posterior flange vs. well-developed posterolingually directed flange in *Kawasphenodon expectatus*). *Navajosphenodon* differs from *Sphenovipera*[15] by the absence of venom grooves on the caniniform successional teeth and an open Meckelian canal anteriorly, instead of closed anteriorly by the contact between the ventral dentary crest and the dorsal dentary crest. Further, *N. sani* differs from *Theretairus antiquus*[48] by having a comparatively much reduced and less pronounced medial curvature of the symphyseal region, two instead of one caniniform successional teeth, and posterior teeth inclined anteriorly in lateral view instead of having an apically projecting cusp with a triangular tooth outline.

**Phylogeny and divergence times.** Results from all analyses, including maximum parsimony, non-clock Bayesian inference (BI), and relaxed morphological clock BI with tip dating, all strongly support the placement of *N. sani* within sphenodontines (Fig. 8 and Supplementary Figs. 1 and 2)—the clade inclusive of the modern tuatara (*Sphenodon punctatus*) besides other species from the Mesozoic and early Paleogene. Recovering *N. sani* within sphenodontines greatly improved the resolution and support of the two major Sphenodontidae clades (Sphenodontinae and Eilenodontinae), besides providing valuable information regarding the skull anatomy in early sphenodontids. As a result, we recover a large number of unambiguous synapomorphies defining Sphenodontinae, which include important skull elements in addition to jaw and dental character states. Unambiguous synapomorphies from the majority rule consensus tree from the morphological clock BI are the presence of nasal foramina; the presence of the anterior process of the quadratojugal; the presence of the posterodorsal process of the coronoid; the presence of anterior caniniform successional teeth; the presence of posterior flanges on the posterior dentary teeth; humeri with an expanded radial condyle.

The overall tree topology and divergence times using relaxed morphological clock BI with tip-dating has the same structure as the equivalent majority rule consensus tree using the same models of evolution of the first published edition of this dataset[19]. However, a major difference concerns the placement of *Sphenotitan* from the Late Triassic of Argentina, now strongly supported as an early evolving Eilenodontinae. *Sphenotitan* shares several anatomical affinities with eilenodontines and has been previously recovered either as the sister taxon to eilenodontines or as an early member of this group—e.g., refs. [3,17,49], or as an early sphenodontid, outside Eilenodontinae and Sphenodontinae[19,50,51]. The previous ambiguous placement of *Sphenotitan* among sphenodontids across various studies is most likely derived from the large amount of missing data constituting most of the sphenodontid fossil species. Until recently, among all sphenodontids, cranial data beyond jaw elements was only available for *Sphenotitan*, the extant *Sphenodon punctatus* (among sphenodontines), and for the Late Cretaceous *Priosphenodon* (among eilenodontines). The recently published *Sphenofontis*, from the Late Jurassic of Germany[21] has just improved the amount of cranial data available for early sphenodontines, but several aspects of its skull morphology remain unknown pending further preparation or CT scanning. Therefore, the paucity of data for

the skull anatomy among early sphenodontines has always imposed a severe constraint on the number of unambiguous synapomorphies recovered for Sphenodontinae, and as a result, the overall number of synapomorphies and support for its sister clade—Eilenodontinae—and internal relationships among early sphenodontids.

The exceptional amount of new data on early sphenodontine anatomy provided by *N. sani* solves the issues above regarding the phylogenetic reconstruction of early sphenodontids. As a result, the phylogenetic placement of *Sphenotitan* is now estimated with strong support (0.94 posterior probability) as the earliest deriving and oldest known eilenodontine. Additionally, node support for Sphenodontidae is also higher than in previous studies (0.8 posterior probability herein vs. <0.6 in ref. [19]). Furthermore, *Sphenotitan* is considerably older than other eilenodontines (representing the only Triassic taxon of the group), and as a result, the time for the most recent common ancestor (MRCA) of all eilenodontines is pushed back into the Late Triassic (median estimate = 213.5 Mya; 95% HPD = 208.6–222.3) and their split from sphenodontines (the Sphenodontidae node) at 223.6 Mya (95% HPD = 212.5–238). This indicates a long gap in the early history of eilenodontines of ~40 Myr, between *Sphenotitan* and the subsequent members of the group to appear in the fossil record in the Middle Jurassic (e.g., *Eilenodon*) (Fig. 8). Finally, the uncertainty around the age estimate for the MRCA of all eilenodontines (95% HPD range = 39.4 Myr in ref. [19]) becomes much reduced after the inclusion of *N. sani* (95% HPD range = 13.7 Myr—Fig. 8 and Supplementary Fig. 3).

**Morphological disparity and morphospace occupation.** The morphospace analysis included nearly all taxa used for the phylogenetic analyses and a subset of characters from the skull and mandibles (see Methods) and indicates a clearly distinct occupation of the cranial morphospace by sphenodontians relative to early lepidosaurs (Fig. 9). Among sphenodontians, most of the morphospace is occupied by members of Clade "A" of ref. [19], composed of *Homeosaurus*, pleurosaurids and saphaeosaurids—this clade was left unnamed because of the uncertainty around the phylogenetic placement of *Homeosaurus* and *Kallimodon* among distinct phylogenetic optimality criteria[19] (and Supplementary Figs. 1–3). The remaining of the morphospace is composed of three smaller clusters comprised of clevosaurids, eilenodontines, and sphenodontines. *Navajosphenodon* occupies the closest position on the morphospace to *Sphenodon*, as expected given the several shared anatomical features between these two taxa.

Some of these results, especially the distinct region of the morphospace occupied by clevosaurids, are similar to a previous analysis focusing on mandibular disparity data[52], with the difference of our sampled South American taxon (*C. brasiliensis* herein) being closer to European taxa than to the North American taxon (*C. bairdi*) (Fig. 9). We also find the South African specimen (*C. sp. SAM*) to be the most distinct among clevosaurids. Importantly, the biggest difference to previous results using mandibular data is on the peripheral placement of sphenodontines, instead of falling closer to the center of the morphospace and close to *Clevosaurus* and other Triassic taxa as in ref. [50].

**Functional morphology.** Despite an overall pattern of decreasing rates of morphological and molecular change throughout sphenodontian evolution[19,24], a large variety of skull shapes can be found throughout the sphenodontian fossil record[20] (see also Figs. 8–10). An important component of this variation can be found in the temporal region of the sphenodontian skull, which houses and protects most of the adductor muscles responsible for the closure of the jaw in all lepidosaurs, including

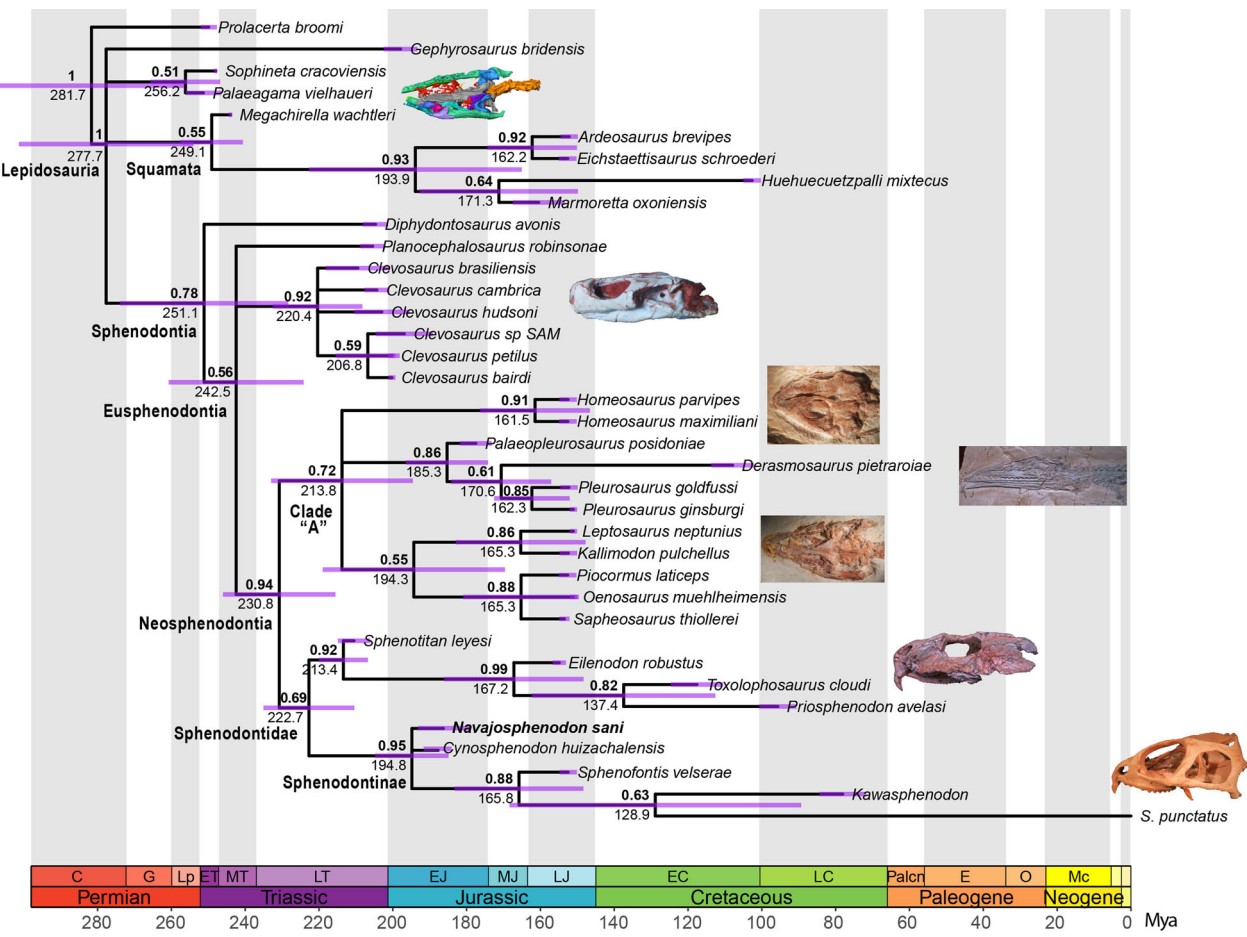

**Fig. 8 Majority rule consensus tree from the relaxed morphological clock Bayesian inference analysis with tip dating.** Results indicate the phylogenetic relationships among sphenodontians highlighting the placement of *N. sani* (in bold), clade posterior probabilities (top node values in bold), and median divergence times (bottom node values). Purple node error bars represent the 95% highest posterior density estimates for divergence times. Skull illustrations (all photos taken by TRS) for each major clade are, from top to bottom, *Megachirella wachtleri* (Squamata), *Clevosaurus brasiliensis* (Clevosauridae), *Homeosaurus maximiliani* (Homeosaurinae), *Pleurosaurus gingsburi* (Pleurosauridae), *Kallimodon pulchellus*, *Priosphenodon avelasi* (Eilenodontinae), and *Sphenodon punctatus* (Sphenodontinae).

*Sphenodon*[47,53,54]. Specifically, the posteroventral process of the jugal forms the lower temporal bar delimiting the lower temporal fenestra, and it undergoes a quite strong degree of variation among sphenodontians, both through ontogeny[7,27,39] and across evolution[20] (Fig. 10). The lower temporal bar restricts the possible size of the external mandibular adductors, and so it has a direct impact on the biting force of lepidosaurs. In fact, the reduction or complete loss of the lower temporal bar in lepidosaurs (most notably its complete loss among the vast majority of squamates) has been considered to be one of their major functional advantages relative to other reptiles[55], as it enabled the expansion of the *M. adductor mandibularis externus superficialis*, which generates significantly more powerful bite forces than in similar sized reptiles with a complete lower temporal bar[56,57].

A reduced lower temporal bar has long been known to be the plesiomorphic condition for sphenodontians, with the complete lower temporal bar in *Sphenodon* representing a reversal to the ancestral diapsid condition[58,59]. Despite the functional advantage of completely losing the lower temporal bar, its reacquisition in *Sphenodon* has been considered an important adaptation for stabilizing the quadrate and reducing the overall stress in the skull during hard biting[58]. Interestingly, *Sphenodon* is not the only sphenodontian with a reversal to a complete lower temporal bar, as it also happened independently in adults of at least some species of *Clevosaurus*[27] (Fig. 10), and possibly in adults of

*Planocephalosaurus*[39]. This reversal also happened (once again, convergently) in two late Cretaceous borioteiioid squamates, *Tyaniusaurus zhengi*[60] and *Polyglyphanodon sternbergi*[31]— the only lizards to have ever re-evolved a completely enclosed lower temporal fenestra[31]. However, differently from the condition observed in most other diapsids, the lower temporal bar in lepidosaurs is almost invariably formed by a posterior elongation of the jugal bone only, which contacts the quadrate (in squamates) or the fused quadrate-quadratojugal (in sphenodontians) by either sutural or ligamentous connections[31,60].

The only known exception to this rule among lepidosaurs is *Sphenodon*, in which the quadratojugal has an anterior extension forming a sutural contact with the jugal and contributing to the enclosure of the lower temporal fenestra, as also observed in archosaurs and other diapsid reptiles with double temporal fenestration. We note, however, that the anterior process of the quadratojugal in *Sphenodon* is shorter compared to that of other diapsids[61]. Yet, *Sphenodon* has been the only lepidosaur known to date with an anatomical reversal to the early diapsid configuration of the temporal region. Therefore, an important question remains on whether the reacquisition of the early diapsid-type temporal region of *Sphenodon* is an oddity of this taxon among all lepidosaurs currently known (either living or extinct), or if it is a general feature of the *Sphenodon* evolutionary branch, but for which we lack informative fossils. As illustrated here by the

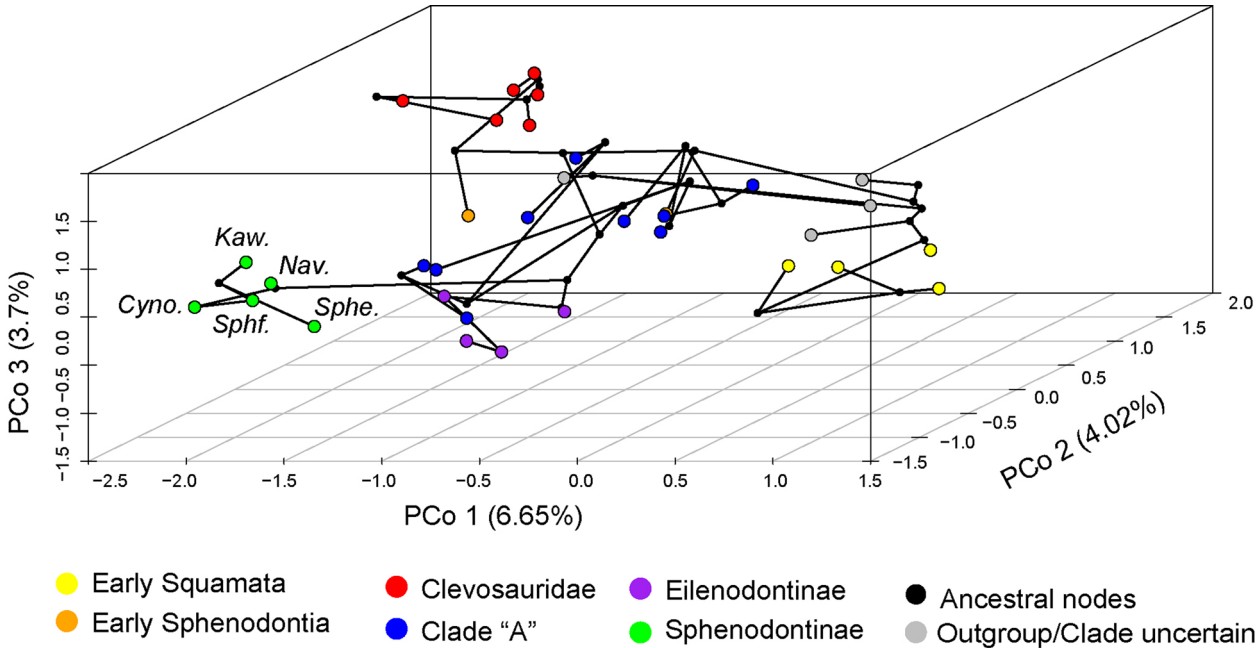

**Fig. 9 Phylomorphospace of early lepidosaurs and sphenodontians using discrete morphological characters.** Clade "A" refers to the clade recovered by the final analysis (relaxed morphological clock Bayesian inference) both here and in Simões, et al.[19], including *Homeosaurus*, pleurosaurids, and saphaeosaurids. For figures with individual taxon names, see Supplementary Fig. 4. Cyno *Cynosphenodon*, Kaw *Kawasphenodon*, Nav *Navajosphenodon*, Sphe. *Sphenodon*, Sphf. *Sphenofontis*.

temporal configuration of *N. sani*, (Fig. 2), it is clear that this condition is not unique to *Sphenodon* only, but most likely a broader feature of all sphenodontines, potentially originating at least as far back as the Early Jurassic. It also directly implies that stabilization of the quadrate and overall stress reduction in the skull during hard biting[58] has much deeper evolutionary origins than previously thought.

## Discussion

The holotype of *N. sani* includes the first nearly complete skull (with a fully articulated skeleton) of any fossil sphenodontine anywhere in the world. Crucially, *N. sani* is also among the oldest known sphenodontines along with (the highly fragmentary) *Cynosphenodon* from the Early-Middle Jurassic of Mexico[14,44], making its cranial data of key importance to recognize skull features diagnosing sphenodontines. As a result, the addition of *N. sani* into the phylogenetic analysis of sphenodontians creates greater stability on sphenodontid relationships. For instance, our results show strong support (0.94 posterior probability) for *Sphenotitan* as the earliest diverging and oldest eilenodontine, extending the origin of that clade into Late Triassic and uncovering a ~40 Myr gap between *Sphenotitan* and later evolving Middle Jurassic eilenodontines. The same applies to sphenodontines, with a long gap of ~50 Myr between *N. sani* in the Early Jurassic to the next unambiguous sphenodontine in the fossil record, *Sphenofontis* from the Late Jurassic of Germany. The latter suggests the early evolution of sphenodontians, although characterized by a richer fossil record compared to squamates[4], still is highly undersampled—especially between the Late Triassic and Late Jurassic.

Our morphospace analysis indicates that clevosaurids occupy a distinct region of the morphospace relative to other sphenodontians, whereas sphenodontines (including *Sphenodon* and *Navajosphenodon*) occupy a peripheral area of the morphospace far away from clevosaurids and Triassic taxa (Fig. 9). Therefore, our results reject the recent hypothesis of *Sphenodon* representing

the conservation of a central morphology on the morphospace of sphenodontians[50]. Instead, sphenodontines and eilenodontines, the two lineages with the youngest fossil records among sphenodontians and the most successful ones during the Cretaceous[12,13,62,63], expanded into new areas of sphenodontian morphospace that were previously unoccupied. Especially in the case of eilenodontines, this enabled the exploration of a novel diet (herbivory) with their highly specialized dentition as a key functional adaptation[45]. Interestingly, however, once invading a new area of the morphospace, both sphenodontines and eilenodontines remained quite restricted on the extent of their occupation of this new morphological zone compared to the space occupied by other sphenodontians (Fig. 9). This is almost certainly driven by the extremely low taxonomic diversity of those two lineages, and of sphenodontians as a whole during the Cretaceous, despite the high abundance of individuals in higher latitudinal environments, especially in South America[62,64].

An important limitation of the morphospace result is the poor fossil record of sphenodontians from the Cretaceous onwards—reflected here and in all available assessments of sphenodontian morphospace[20,50,52]. It is possible that this poor fossil record is a representation of true low taxonomic diversity and not simply a collection bias, at least in the northern hemisphere where collection efforts have historically been more intense than in southern continents[65]. But growing research effort, especially in South America[65], has the potential to shed light on this matter in the near future.

The new data provided by *N. sani* highlights the remarkable convergence on the configuration of the temporal region between sphenodontines and early diapsid reptiles. In particular, the new species demonstrates that the complete lower temporal bar formed by the jugal and an anterior projection of the quadratojugal is not exclusive to *Sphenodon* and was already present among the earliest members of sphenodontines during the Early Jurassic at ~190 Mya. It suggests that the functional adaptive value of this particular type of lower temporal bar evolved early to reduce skull stress during forceful biting[31,58] and may have played

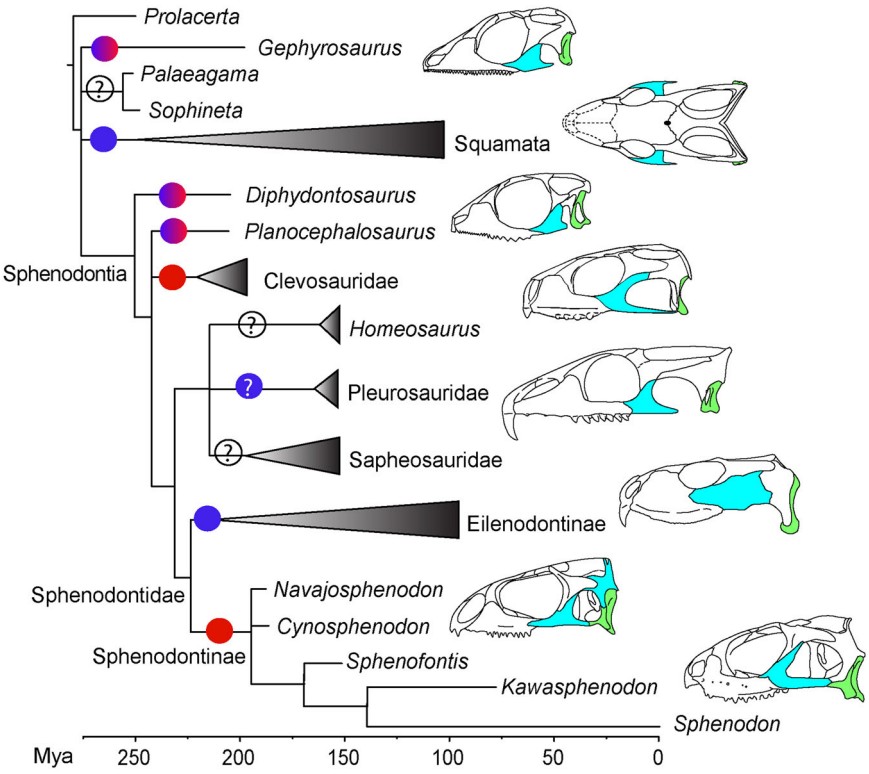

**Fig. 10 Evolutionary changes on the temporal region of sphenodontians.** Many early-evolving lepidosaurs retain a single temporal fenestration as the jugal (cyan) does not contact the quadrate/quadratojugal (green) posteriorly. A full development of the lower temporal bar and double temporal fenestration evolved independently at least twice in sphenodontians, once among clevosaurids and once in sphenodontines—an adaptation for stabilizing the quadrate and reducing overall stress in the skull during hard biting[58]. The latter is unique among lepidosaurs by including contributions from both the jugal and quadratojugal—a morphology convergent with many non-lepidosaurian early diapsid reptiles. Red circle, complete lower temporal bar; blue circle: incomplete lower temporal bar; blue-red gradient circle, lower temporal bar incomplete in juveniles but complete in adults. Skull drawings from top to bottom: *Gephyrosaurus* (drawn by TRS based on ref. [20]), *Megachirella* (re-drawn by TRS from ref. [33]), *Diphydontosaurus* (drawn by TRS based on ref. [20]), *Clevosaurus* (drawn by TRS based on ref. [20]), *Palaeopleurosaurus* (drawn by TRS based on ref. [20]), *Navajosphenodon* (drawn by A. Brum), and *Sphenodon* (drawn by TRS based on MCZ R4702).

an important adaptive role during the early divergence of sphenodontines and the acquisition of the tuatara skull-type among sphenodontians.

The similar configuration of the temporal region is not the only remarkable anatomical similarity between *Navajosphenodon* and *Sphenodon*. In fact, there is a remarkable number of similarities in their skull configuration relative to other sphenodontians, which also drives them close together in the sphenodontian morphospace (Fig. 9), despite these two genera being separated by 190Myr (Fig. 8). Some few exceptions include differences on the anterior end of the snout (e.g., shape of the premaxilla and multiple premaxillary teeth) and the presence of a midventral crest on the vertebrae of *N. sani*. However, individually recognizable premaxillary teeth instead of a single fused premaxillary tooth were also detected in a recently described sphenodontine (*Sphenofontis*)[21], thus strongly indicating this was the plesiomorphic condition for sphenodontines, and persisting at least until the end of the Jurassic. Additionally, the morphological changes detected on the upper and lower jaws of *N. sani* during its ontogenetic development are also very similar to the ontogenetic changes observed in *Sphenodon* (Fig. 7), with the possible exception of the retention of alternating teeth in the dentary on older individuals of *N. sani*.

Although the morphological differences between *Sphenodon* and *Navajosphenodon* are more than enough to clearly differentiate the two taxa taxonomically, those are very minor from a deep time evolutionary perspective considering the enormous time span separating the two. Even in the context of sphenodontian evolution, within a much smaller time span—between the Late Triassic and Late Jurassic (220–145 Mya, or 75 Myr)—we see the evolution of all other sphenodontian morphotypes, including a large diversity of skull shapes[20] (Figs. 8–10). These anatomical similarities between *Navajosphenodon* and *Sphenodon*, most notably on the structure of the temporal region of the skull, along with the recently detected exceptionally low rates of morphological and molecular evolution on the evolutionary branches leading to *Sphenodon*[19,24], suggest deep time conservation of the tuatara-lineage morphotype, at least since the Early Jurassic.

The sphenodontian fossil record is marked by numerous taxa mostly represented by fragmentary jaw elements that limit our ability to reconstruct some key evolutionary patterns in sphenodontian evolution. Perhaps no other group characterizes this issue better than sphenodontines, the group that includes *Sphenodon punctatus*, the New Zealand tuatara—the sole living representative of sphenodontians. Although the sphenodontine fossil record dates back to the Early Jurassic[14] (and this study) and it has been estimated to have originated at ~190 Mya[19] (and this study), it is mostly known from isolated jaws and vertebrae, with almost no fossil record for the entire Cenozoic (66 Mya–present). This has resulted in great phylogenetic instability regarding the placement of *Sphenodon* and its closest allies among sphenodontians in several previous studies[50,51,63,66,67], besides preventing our understanding of the origin of some of its key particular features, such as the reacquisition of an early diapsid-like temporal region

that is unique among lepidosaurs. Here we describe the first sphenodontine from North America (Arizona, USA) that is represented by an almost fully preserved skeleton containing a partially articulated skull and postcranium (Figs. 1–7), named *N. sani*, which helps to stabilize and improve phylogenetic support for the relationships among sphenodontids. This has a direct impact on the placement of other key taxa, such as recognizing *Sphenotitan* as the oldest and earliest evolving eilenodontines with strong phylogenetic support (Fig. 8). As a consequence, divergence time estimates using relaxed morphological clocks for the origin of eilenodontines are shifted almost 40 Myr into the past, indicating a long sampling gap in the early history of that group.

The skeleton of *N. sani* shows a large number of similarities with the modern tuatara *S. punctatus*, clustering them closely together in the morphospace of sphenodontians and early lepidosaurs (Fig. 9). These two genera, separated by 190 Myr of evolutionary history, show a similar or even greater degree of morphological similarity than that of congeneric sphenodontian species, such as all species within *Clevosaurus* or the two species of *Pleurosaurus*. Greater morphological disparity is also achieved by other sphenodontians living between the Triassic and Late Jurassic (a time span of ca. 75 Myr), in comparison to sphenodontines during a much longer time span (190 Myr). Combined with recent data indicating slow rates of both morphological[19] and genomic[24] evolution leading to *Sphenodon*, this leads us to suggest the hypothesis that sphenodontines underwent a remarkable degree of stabilizing selection and morphological conservatism throughout their extremely long evolutionary history, since at least the Early Jurassic. Skull features previously thought to be exclusive to *Sphenodon* among all lepidosaurs, such as the reacquisition of an early diapsid-type double temporal fenestration of the skull, had already been developed much earlier in the history of its lineage than previously thought. Importantly, the latter suggests that functional adaptations for hard-biting during prey capture enabled by a complete lower temporal bar[58] may have played an important role in the origin of sphenodontines, and was a successful adaptation (possibly) sustained for at least 190 Myr, rather than being restricted to its single extant representative.

## Methods

**Institutional abbreviations**. MCZ, Museum of Comparative Zoology, Harvard University—Cambridge, MA, USA; MNA, Museum of Northern Arizona—Flagstaff, AZ, USA.

**Samples**. The specimens described here were collected during the 1982–1983 field seasons led by Dr. Farish A. Jenkins Jr. and remained undescribed for decades in the vertebrate paleontology collections of the Museum of Comparative Zoology, Harvard University. *N. sani* is represented by 15 specimens, consisting mostly of dentary and maxillary elements. Among those, the most complete specimen was designated as the holotype, consisting of one fully articulated individual (MNA.V.12442, Figs. 1–7). Due to the diminutive size of most specimens and the delicate nature of the bones most specimens were not mechanically prepared.

**High-resolution micro-CT scanning**. The holotype (MNA.V.12442) was micro-computed tomography scanned in Harvard University's Center for Nanoscale Systems. The whole skeleton of the holotype (MNA.V.12442) was scanned using an X-Tek Micro-CT scanner (model XRA-022 HMXST225) at 90 kV, 170 μA, and with a voxel size of 41.97 μm. To enhance anatomical resolution, the skull of MNA.V.12442 was scanned using an Xradia Versa 620 Zeiss micro-CT scanner at 90 kV, 134 μA, and with a voxel size of 11.5 μm. The resulting x-rays were reconstructed as a sliced image sequence after re-centering and ring artifact correction with VGSTUDIO MAX CT Reconstruction Module. The reconstructed slices were further enhanced in contrast and cropped using ImageJ. Bones were segmented and 3D rendered in Dragonfly 4.0 (Object Research Systems, available at http://www.theobjects.com/dragonfly) and MIMICS v.22 (Materalise, Belgium). The final segmented objects were all exported as individual mesh files in STL format and are available as online supplementary material in MorphoSource (see Data Availability).

**Morphological dataset**. We assessed the phylogenetic placement of *N. sani* in the recently published sphenodontian phylogenetic dataset of Simões, et al.[19], which includes several new and heavily revised phylogenetic characters following standard guidelines for dataset construction—e.g., refs.[68–70]—and comprising all major groups of sphenodontians—mostly based on direct personal observation of the sampled species. The updated final dataset includes 36 taxa utilized for the analyses and 131 characters—dataset freely available online at Harvard's Dataverse[71] (see "Data availability").

**Maximum parsimony phylogenetic analysis**. Analyses were conducted in TNT 1.1[72] with all characters unordered. All heuristic searches were done under equal weights and consisted of 1000 rounds of random addition sequence (RAS) of taxa followed by Tree Bisection Reconnection (TBR) branch swapping, holding 90 trees per replication and collapsing branches of zero length after tree search. The resulting trees were used as starting trees for a final round of TBR branch swapping.

**Non-clock and relaxed morphological clock Bayesian inference analyses**. In a recent study, we conducted a thorough examination of nearly all possible combinations of models of character evolution, clock models, and tree models on the same core dataset used here[19]. Therefore, for the present analyses we implemented the best performing model combinations reported in[19] for both non-clock and morphological clock Bayesian inference (BI)—we refer the reader to Simões et al.[19] for further details. Our morphological model for non-clock BI thus includes the Mkv morphological model of evolution with assortment bias correction for the absence of invariable characters[73]; gamma probability distribution for among character rate variation; symmetric distribution of character state frequencies; all characters unordered (equal transition probabilities among all states).

For morphological clock BI, the morphological model of evolution is the same as in the non-clock BI analyses. Additionally, the clock and tree models include the following best set of model parameters:[19] the TK02 continuous autocorrelated clock model;[74] maximizing diversity without sampled ancestors (all fossils as tips only)[19,75]; the skyline fossilized birth–death model (SFBD) with two predefined time slices (three in total considering the time after the last sampled fossil, or cutoff time $x_{cut}$)[75].

Analyses were conducted using Mr. Bayes v. 3.2.7a[76] using the CIPRES Science Gateway v.3.3[77]. Convergence of independent runs was assessed using: average standard deviation of split frequencies (ASDSF ~0.01), potential scale reduction factors [PSRF ≈1 for all parameters], and effective sample size (ESS) for each parameter greater than 200, and analyzed using Tracer v. 1.7.1[78].

**Morphospace analysis—dataset adaptation**. Using a morphological phylogenetic dataset for the analysis of morphospace occupation and morphological disparity requires dataset adaptations. The most important of all being the reduction in the total amount of missing data[79]. To reduce the negative impact of missing data, we removed all characters with >40% of missing data from the dataset, which resulted in a total of 28.6% missing data on the final dataset. This value is within the maximum threshold of 25–30% of missing data which enables reliable disparity estimates[79–81]. Additionally, inapplicable characters are a big conceptual problem to constructing a morphospace. Taxa with inapplicable characters will have their placement enforced upon a space they do not reside in (which is conceptually very different from missing data—when they reside in that space, but we currently lack data to place them)[79]. After deletion of characters with high amounts of missing data, most remaining characters had very little amounts of inapplicable scores across taxa. Excluding characters with at least one inapplicable score would highly reduce the number of sampled characters, and so we only deleted characters with more than 5% of inapplicable scorings. Polymorphisms were converted into NA scores (treated as "?" during analyses), following previous recommendations and based on the reasoning above[79,82]. Autapomorphies, if unevenly sampled across taxa, may also contribute to bias distance matrices. However, if autapomorphies are uniformly distributed across terminal taxa, then their overall effect is to increase overall pairwise distance between terminal taxa uniformly, therefore not creating biasing the interpretation of the data[83]. In the present dataset, there are some directly observed autapomorphies, which were already excluded during the removal of characters with large amounts of missing data (see below). Having no remaining autapomorphies in the dataset is another way of having a uniform distribution of autapomorphies, guaranteeing that no taxon will have additional dimensions separating it from other taxa in the dissimilarity matrix and morphospace ordination procedure.

**Intertaxon distance matrix and ordination analysis**. The procedures above resulted in a final reduced matrix of 36 taxa and 43 characters—a sufficient sampling to provide reasonable estimates of disparity[84]. This data subset was used to construct a morphospace for all sampled taxa.

Using the reduced datasets and the time-calibrated trees, we constructed an intertaxon distance matrix D and an ordination matrix using principal coordinate analysis (PCoA—or classical multidimensional scaling). We implemented MORD as our method of estimating pairwise taxon distances for the distance matrix D and the subsequent ordination matrix, made available through the Claddis R package[81],

implementing Cailliez's correction for negative eigenvalues. Additionally, we increased our sample size by including internal nodes using ancestral state reconstructions through the recently developed pre-OASE1 method[85]. This procedure provides a much better approximation of the true morphospace when compared to methods to reconstruct ancestral nodes in most previous disparity studies using ancestral state reconstructions[85]. Claddis removes taxa with high amounts of missing data for which it could not meaningfully estimate pairwise taxon distances. For the present dataset, it excluded the following taxa: *Ardeosaurus brevipes*, *Piocormus laticeps*, *Ardeosaurus brevipes*+*Eichstaettisaurus schroederi*, *Piocormus laticeps*+*Sapheosaurus thiollerei*, *Homeosaurus maximiliani* +*Homeosaurus parvipes*, and *Homeosaurus maximiliani*. PCoA were performed also using Claddis in R[81].

Previous analyses of sphenodontian morphological disparity and morphospace occupation were based on 2D morphometric characters obtained from the mandibles[50,52] or from drawings of reconstructed fossil skulls[20]. Despite the necessary reduction in the number of characters used for assessing the morphospace of sphenodontians herein to avoid spurious results driven by large amounts of missing data, the final number of variables and taxon sampling assessed herein are similar in scale to those previous studies. However, as we combine data from both mandibles (including dentition) and the skull, it is difficult to assess whether differences between our results and previous ones stem from using discrete instead of morphometric variables or by integrating information from both cranial regions.

**Nomenclatural acts**. This published Article and the nomenclatural acts it contains have been registered in ZooBank, the proposed online registration system for the International Code of Zoological Nomenclature. The ZooBank Life Science Identifiers (LSIDs) can be resolved, and the associated information can be viewed through any standard web browser by appending the Life Science Identifier to the prefix http://zoobank.org/. The LSIDs for this publication are urn:lsid:zoobank.org:pub:457AC973-DC8A-477F-8DF6-07DD785AC57C; urn:lsid:zoobank.org:act:00669666-B46C-4CF5-88A1-2BE57FAA1B08; urn:lsid:zoobank.org:act:6DF40030-2354-4A26-AECA-CF36BBEA36F4.

**Reporting summary**. Further information on research design is available in the Nature Research Reporting Summary linked to this article.

## Data availability
All morphological datasets used for phylogenetic and morphospace analyses, along with the output files generated and R scripts, are available online as Supplementary Data files at Harvard's Dataverse Repository[71]: https://doi.org/10.7910/DVN/KZQZ2X. All CT scanned data generated, and 3D object files are available at MorphoSource: https://www.morphosource.org/concern/biological_specimens/000391918?locale=en. All MorphoSource DOIs for each individual digitized element are provided in Supplementary Table 2.

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

## Acknowledgements

T.R.S. was supported by an Alexander Agassiz Postdoctoral Fellowship and an Ernest Mayr Travel Grant (Museum of Comparative Zoology, Harvard University). This work was also supported by funds made available to S.E.P. by Harvard University. We thank the MCZ Vertebrate Paleontology collection staff for their assistance, C. Byrd, E. Biedron, C. Green, and J. Cundiff. We also thank A. Brum for assistance with one of the drawings in Fig. 9. A version of this work was submitted by G.K.-B. in partial fulfilment of her undergraduate senior thesis at Boston College and conducted in the Pierce laboratory. The authors extend a special thanks to the Navajo Nation, and F.A. Jenkins, Jr. and his field crew for discovering, collecting, and preparing the fossils described in this paper. The fossils were collected under permit No. 82-AZ-139, in direct charge of F.A. Jenkins, Jr. and C.R. Schaff and the holotype specimen was discovered by F.A. Jenkins, Jr. in 1982. Fieldwork on the Navajo Nation was conducted under a permit from the Navajo Nation Minerals Department. Any persons wishing to conduct geologic investigations on the Navajo Nation must first apply for, and receive, a permit from the Navajo Nation Minerals Department, P.O. Box 1910, Window Rock, Arizona 86515, and telephone # (928) 871-6588.

## Author contributions

S.E.P. and T.R.S. conceived and designed the study. T.R.S. led on the project, manuscript writing, conducted analyses, and created the figures. S.E.P, T.R.S., and G.K.-B. conducted CT scanning and G.K.-B. segmented the CT scan data. All authors contributed to discussions and manuscript editing and approved the final version of the manuscript.

## Competing interests

The authors declare no competing interests.
