## [Transparent Peer Review File · Communications Biology]

Reviewers' comments:

Reviewer #1 (Remarks to the Author):

This paper presents a very interesting new sphenodontine reptile from the Early Jurassic of North America, of which not only the holotype but also several different growth stages are preserved. I think the authors made a good case for their claim of "ontogenetic conservatism", although they remain only qualitative-descriptive in their assessment and do not perform a somewhat more quantitative (e.g. 2D geometric-morphometric) approach.

I have one critical comment on the discussion of the temporal arcade: yes, the quadratojugal in *Sphenodon* does have a small anterior projection, but it is still much shorter than the typical condition seen in early diapsids. I myself have been discussing this issue at length in an earlier paper of mine (Müller, J. 2003: Early loss and multiple return of the lower temporal arcade in diapsid reptiles. *Naturwissenschaften* 90, 473–476).

Some minor issues/typos:

Line 248: "dorsum sellae" not "dorsum sella" (the latter is grammatically wrong).

Line 250: "abducens canal" not "abducens canal".

Line 383: "well-defined and"

Johannes Müller

Reviewer #2 (Remarks to the Author):

The study describes a series of specimens new to science from the Early Jurassic Kayenta Fm of North America. The best of these includes a relatively complete skull and postcranium, which is an exceedingly rare and exciting discovery for the terrestrial forms of this group. A phylogenetic analysis based on a matrix whose core was published previously, supports the specimens as sphenodontine sphenodontians – a nominally confusing position to be, but a position shared with the only living sphenodontian, the tuatara (*Sphenodon punctatus*). The implication here is that the body plan of the living tuatara was in place at least by the Early Jurassic and thus the sole living species reflects an extended period of evolutionary conservation. This conclusion is supported by a morphospace analysis whose primary empirical structure was also published previously. The conservative nature of *Sphenodon* features is not limited to the adult form but extends to basic growth trajectories; this finding is not necessarily surprising, but it is satisfying to establish with some rigor. The study thus weighs in on the debate of whether the living tuatara represents a "living fossil" (the authors, probably to their credit, don't employ this terminology); this is not terribly important in-and-of-itself but considering that the tuatara is often used as a proxy for the basal reptile condition in broad comparative studies, it is also not without implication.

All-in-all, the study is an important one for the macroevolution of reptiles. I have a short series of comments/suggestions/requests.

1. The lasting contribution of this study will be the fossils themselves, so the images must be better. The 3D reconstructions are perhaps fine for some elements but inadequate for the skull and most of the cranial partitions. You simply cannot see the details. If the specimen is actually this important, then it would seemingly be worthy of some line drawings alongside the reconstructions.

2. Reconstruction of the skull. I made this a separate line item for emphasis. I realize that there is some danger in providing a line drawing of the reconstructed skull because subsequent authors will rely too heavily on that interpretation rather than the actual specimen(s). But readers also want to

better understand your interpretations as they are conveyed in the descriptive text. Please add.

3. In general, the writing is fairly sloppy and filled with grammatical errors, inconsistencies, and odd, non-sensical phrasing. I started to list all the little issues but began to wonder why I was doing the authors work for them, so I stopped. These could be easily fixed if the authors would just do a thorough job of reading through each line of text critically. Some examples from early in the manuscript include:

- a. The first sentence is misleading, at least if you are working within the phylogenetic system. Since sphenodontians and squamates are sister taxa, their age would be the same as you actually point out in the next sentence. So, what is it exactly that you mean? Please be more specific and exact in your language.
- b. In the abstract, you use "US" and in the main text, you use "USA" Just be consistent.
- c. Line 38: This sentence is oddly worded. What do you mean by "most specie-rich group of lepidosaur only in the Cretaceous"?
- d. Line 50: "Most notably to this patchy fossil record..." I don't know what this means.
- e. Line 61: It's odd to refer to the tuatara as a model organism. It certainly is not that in the sense in which that term is generally applied. *E. coli* is a model organism, as is *Mus musculus* – not *Sphenodon*. It is often used a representative "early/plesiomorphic" reptile, for better or worse.
- f. Line 63: reconsider the term "leading to" – these transformations produced the modern condition. Leading to seems rooted in teleological thinking, even though I know that isn't what you mean.
- g. Line 67: What are distinct phylogenetic methods? As opposed to vague phylogenetic methods?
- h. Line 183: "in much better state of preservation"
- i. Line 190: "very" is a weak word. Doesn't really add anything here.

4. I'm sure that you justify the use of Sphenodontia in the 2020 paper, but it's unfortunate because Rhynchocephalia is so easier for communication. I'm constantly double checking to see if you are talking about the whole radiation or the specific clade of which *Sphenodon* is a part – especially in the less formal colloquial usage (sphenodontians versus sphenodontines, ugh)

5. You state the premaxillae lack a posterior maxillary process (apart from the posterodorsal process) and that this is an unambiguous autapomorphy for the group. This needs more explanation and a figure that shows what you mean more clearly. Does *Sphenodon* have this process? Looking at figure 2 of Jones et al. 2009 – it certainly looks as if it also lacks any such process. Please explain.

6. You mention that there is little evidence that the frontals made a contribution to the orbital margin – does this mean there was prefrontal-postfrontal contact? You don't mention direct evidence for this.

7. Restrict the citation of personal observation to those that are truly novel. A lot of them seem to be conveying that you agree with previous publications, which is fine but not worthy of individual citations.

8. Line 241: please add a citation for the point that the first ceratobranchial is usually mineralized in lepidosaurs – unless that is a pers. Ob., then cite it as such.

9. You note in the caption of Fig. 10 that "A full development of the lower temporal bar and double fenestration evolved independently at least twice..." This is misleading. A fully formed lower temporal bar may have evolved twice but the double fenestration certainly did not. The ossified lower temporal bar is a feature distinct from the lower temporal bar that it bounds. Squamates to my knowledge are universally accepted as having a lower temporal fenestra despite the fact that their quadratomaxillary ligament fails to fully ossify.

Reviewers' comments:

Reviewer #1 (Remarks to the Author):

This paper presents a very interesting new sphenodontine reptile from the Early Jurassic of North America, of which not only the holotype but also several different growth stages are preserved.

I think the authors made a good case for their claim of “ontogenetic conservatism”, although they remain only qualitative-descriptive in their assessment and do not perform a somewhat more quantitative (e.g. 2D geometric-morphometric) approach.

Response: we thank the referee for their feedback and we are happy to learn that they agree with our observations of ontogenetic conservatism. With reference to quantitative approaches to the assessment of morphological (shape) variation, we are strong supporters of this approach to address several questions in evolutionary biology—e.g., (Pierce et al., 2008; Pierce et al., 2009; Jones et al., 2018; Dickson et al., 2020). However, in this particular instance, the amount of comparable elements available across ontogenetic stages is too low (< 10 relatively completely dentaries preserved in situ at various orientations) to provide reasonable sample sizes for substantial multivariate statistical analyses derived from geometric morphometrics. Alternatively, one could try to place the data points from the fossil species in an ontogenetic morphospace of the modern *Sphenodon*. However, there is no published ontogenetic series of *Sphenodon* nor data available at our disposal to construct such morphospace. Finally, some of the ontogenetic variation that we observe include changes in the tissues of tooth attachment that would not be able to be captured by linear or geometric morphometrics.

I have one critical comment on the discussion of the temporal arcade: yes, the quadratojugal in *Sphenodon* does have a small anterior projection, but it is still much shorter than the typical condition seen in early diapsids. I myself have been discussing this issue at length in an earlier paper of mine (Müller, J. 2003: Early loss and multiple return of the lower temporal arcade in diapsid reptiles. *Naturwissenschaften* 90, 473–476).

Response: We agree that the anterior process of the quadratojugal in *Sphenodon* is reduced compared to that of other diapsids, when it is present. We now make mention of this in the main text and include the reference above: lines 606-607.

Some minor issues/typos:

Line 248: “dorsum sellae” not “dorsum sella” (the latter is grammatically wrong).

Line 250: “abducens canal” not “abduscens canal”.

Line 383: “well-defined and”

Response: Done. Thank you for the editorial suggestions.

Johannes Müller

Reviewer #2 (Remarks to the Author):

The study describes a series of specimens new to science from the Early Jurassic Kayenta Fm of North America. The best of these includes a relatively complete skull and postcranium, which is an exceedingly rare and exciting discovery for the terrestrial forms of this group. A phylogenetic analysis based on a matrix whose core was published previously, supports the specimens as sphenodontine sphenodontians – a nominally confusing position to be, but a position shared with the only living sphenodontian, the tuatara (*Sphenodon punctatus*). The implication here is that the body plan of the living tuatara was in place at least by the Early Jurassic and thus the sole living species reflects an extended period of evolutionary conservation. This conclusion is supported by a morphospace analysis whose primary empirical structure was also published previously. The conservative nature of *Sphenodon* features is not limited to the adult form but extends to basic growth trajectories; this finding is not necessarily surprising, but it is satisfying to establish with some rigor.

Response: We thank the referee for their detailed assessment of our manuscript. However, we respectfully disagree that our findings on the conservative nature of the *Sphenodon* body plan and ontogeny is expected. As far as we are aware, and based on a thorough literature search, no one has ever suggested or predicted that the *Sphenodon* body plan has been conserved for nearly 200 million years. We believe this finding to be a point of substantial impact and a novel contribution from the present study.

The study thus weighs in on the debate of whether the living tuatara represents a “living fossil” (the authors, probably to their credit, don’t employ this terminology); this is not terribly important in-and-of-itself but considering that the tuatara is often used as a proxy for the basal reptile condition in broad comparative studies, it is also not without implication.

Response: Indeed. We also note that the living fossil terminology debate is not central to any of the core results and discussions in this manuscript. However, we do make mention of it in light of recent literature raising this topic (as cited in the relevant sections of the manuscript).

All-in-all, the study is an important one for the macroevolution of reptiles. I have a short series of comments/suggestions/requests.

1. The lasting contribution of this study will be the fossils themselves, so the images must be better. The 3D reconstructions are perhaps fine for some elements but inadequate for the skull and most of the cranial partitions. You simply cannot see the details. If the specimen is actually this important, then it would seemingly be worthy of some line drawings alongside the reconstructions.

Response: We respectfully disagree that including line drawings would improve the quality of the anatomical interpretation: 1) Hand drawings are substantially more prone to errors compared to individual bone segmentations and illustrations, such as those provided here. This is reinforced here by the referee’s comment about the posterior process of the premaxilla (see below). To reduce these errors, most studies utilizing high-resolution CT scanning don’t rely on hand drawings anymore; and 2) Quite importantly, all readers will be able to download every single element of the skull and most of postcranium as high-resolution 3D objects from our files deposited on MorphoSource. Any researcher interested in the anatomy of the new species will be able to have a three-dimensional view at an extremely high resolution of ~11 microns, which

is substantially more detailed than most fossil CT scans available for most reptile groups and of far greater scientific value than hand drawings.

We also would like to note here that image quality in initial submissions for Nature research journals tend to be much lower resolution because they are embedded in the manuscript text file. Final figures used for publication will be taken from raw figure files during the editorial stage, which ensures exceptionally high-resolution images will accompany the paper.

2. Reconstruction of the skull. I made this a separate line item for emphasis. I realize that there is some danger in providing a line drawing of the reconstructed skull because subsequent authors will rely too heavily on that interpretation rather than the actual specimen(s). But readers also want to better understand your interpretations as they are conveyed in the descriptive text. Please add.

Response: Such a line drawing reconstruction was already present in figure 10 of the original submission (image just next to the name *Navajosphenodon*, and identified in the figure caption). However, as indicated above, hand drawing reconstructions can be extremely deceiving, and we utilized it here merely for illustrative purposes of our perceived aspect of the skull in early sphenodontines, not as a source of anatomical data for downstream analyses.

3. In general, the writing is fairly sloppy and filled with grammatical errors, inconsistencies, and odd, non-sensical phrasing. I started to list all the little issues but began to wonder why I was doing the authors work for them, so I stopped. These could be easily fixed if the authors would just do a thorough job of reading through each line of text critically. Some examples from early in the manuscript include:

Response: we respectfully, and strongly, disagree with the referee in characterizing our writing as “fairly sloppy” and such a view was not expressed by referee 1. Nonetheless, we have reviewed the text thoroughly to catch any lingering grammatical errors.

a. The first sentence is misleading, at least if you are working within the phylogenetic system. Since sphenodontians and squamates are sister taxa, their age would be the same as you actually point out in the next sentence. So, what is it exactly that you mean? Please be more specific and exact in your language.

Response: from our original text: “*Sphenodontians are one of the longest living lineages of extant reptiles, with a fossil record of at least 230 million years*”. We make reference to the fossil record (the oldest known fossil), not the age of the group itself. Later in the sentence we talk about when both groups diverged, not the fossil record: “*recent morphological and molecular clock estimates suggesting their split from their closest relatives—squamates (lizards and snakes)—during the Late Permian at about 259Mya*”. We are unsure how to make this clearer.

b. In the abstract, you use “US” and in the main text, you use “USA” Just be consistent.

Response: Done.

c. Line 38: This sentence is oddly worded. What do you mean by “most specie-rich group of lepidosaur only in the Cretaceous”?

Response: Species-rich literally means richness of species, which is a common ecological variable, and a synonym to speciose.

d. Line 50: “Most notably to this patchy fossil record...” I don’t know what this means.

Response: Reworded, also to reflect a recent publication.

e. Line 61: It’s odd to refer to the tuatara as a model organism. It certainly is not that in the sense in which that term is generally applied. *E. coli* is a model organism, as is *Mus musculus* – not *Sphenodon*. It is often used a representative “early/plesiomorphic” reptile, for better or worse.

Response: we do not consider this a writing issue, but a specialist topic of debate. In that sense, considering a species as a model organism is somewhat subjective. Yet, taking the referee’s comment into account, we removed this terminology from the text.

f. Line 63: reconsider the term “leading to” – these transformations produced the modern condition. Leading to seems rooted in teleological thinking, even though I know that isn’t what you mean.

Response: Done. “...evolutionary changes between early sphenodontians and *Sphenodon*.”

g. Line 67: What are distinct phylogenetic methods? As opposed to vague phylogenetic methods?

Response: Distinct = different (<https://www.merriam-webster.com/thesaurus/distinct>). Considering this sentence (“Using high resolution micro-CT scanning and distinct phylogenetic methods...”), it should be clear we are referring to different phylogenetic methods.

h. Line 183: “in much better state of preservation”

Response: the referee merely pasted here an extract of our text without indicating what exactly they would like us to correct.

i. Line 190: “very” is a weak word. Doesn’t really add anything here.

Response: We do not understand what the referee means by a “weak” word. “Very” is an adverb of intensity that is commonly utilized in anatomical descriptions to discern anatomical components based on their size, especially in a comparative context, as done herein: “*The anteroventral process is very elongate, being much longer than the posteroventral process...*”.

4. I’m sure that you justify the use of *Sphenodontia* in the 2020 paper, but it’s unfortunate

because Rhynchocephalia is so easier for communication. I'm constantly double checking to see if you are talking about the whole radiation or the specific clade of which Sphenodon is a part – especially in the less formal colloquial usage (sphenodontians versus sphenodontines, ugh)

Response: As indicated by the referee, there are several reasons for the extinction of the term Rhynchocephalia and its replacement by Sphenodontia that were detailed in our previous paper (Simões et al., 2020). This is why we refer the reader to that study regarding this terminology, and we also take the opportunity to invite the referee to read our detailed reasoning in our 2020 paper. We also would like to remind the referee that we were not the proponent of those names in our previous study in 2020. We merely formalized – phylogenetically - the definition of already existing taxonomic names.

Importantly, this terminology is not a result or a central point of the present manuscript, and so we are unsure of the reasons to address it here.

5. You state the premaxillae lack a posterior maxillary process (apart from the posterodorsal process) and that this is an unambiguous autapomorphy for the group. This needs more explanation and a figure that shows what you mean more clearly. Does Sphenodon have this process? Looking at figure 2 of Jones et al. 2009 – it certainly looks as if it also lacks any such process. Please explain. Response: a figure was already provided in the original submission to illustrate this and other features of the premaxilla—the 3D rendering of the CT scan data from the premaxilla in lateral and posterior views (Figs. 2a,b). This figure has now been called explicitly in this part of the text.

In regards to the absence of the posterior maxillary process of the premaxilla in *Sphenodon*, unfortunately the illustration referenced by the referee is only a drawing in lateral view that does not show this process, which is mostly medial to the anterior process of the maxilla. See figure below from our CT scan data from *Sphenodon*:

6. You mention that there is little evidence that the frontals made a contribution to the orbital margin – does this mean there was prefrontal-postfrontal contact? You don't mention direct evidence for this.

Response: No. We mentioned that they contributed **little** to the orbital margin, not zero. Thus, they still made some contribution to the orbital margin, which logically implies that pre- and postfrontals were not contacting each other.

7. Restrict the citation of personal observation to those that are truly novel. A lot of them seem to be conveying that you agree with previous publications, which is fine but not worthy of individual citations.

Response: in comparative anatomy, especially in the fossil record, personal observation of traits have a prerogative over any published observation, especially the older literature in which image quality was far inferior compared to what is available today. For transparency, and to acknowledge that we have made first-hand observations, we believe it is important to note when we have personally observed an anatomical feature.

8. Line 241: please add a citation for the point that the first ceratobranchial is usually mineralized in lepidosaurs – unless that is a pers. Ob., then cite it as such.

Response: Susan Evans (pers. Comm., 2011)— Added to the manuscript. Line 239.

9. You note in the caption of Fig. 10 that “A full development of the lower temporal bar and double fenestration evolved independently at least twice...” This is misleading. A fully formed lower temporal bar may have evolved twice but the double fenestration certainly did not. The ossified lower temporal bar is a feature distinct from the lower temporal bar that it bounds. Squamates to my knowledge are universally accepted as having a lower temporal fenestra despite the fact that their quadratomaxillary ligament fails to fully ossify.

Response: In principle, we agree with this observation. Unfortunately, unossified lower temporal bars are rarely, if ever, preserved in the fossil record. For this reason, the entire history of research surrounding the evolution of temporal fenestration in reptiles has always used the *ossified* temporal bars as a universal marker for temporal fenestration. So, our terminology and criteria for defining temporal fenestrae follows that of several previous studies, including the one mentioned by referee 1 (see above). E.g.:

(Evans and Jones, 2010): “*Thus the first lepidosauromorphs inherited a skull in which the ventral margin of the lower temporal fenestra was open, the quadratojugal was small, and the jugal lacked a posterior process (Fig. 2.3).*” → Making reference to specimens with incomplete ossification of the lower temporal bar among early lepidosaurs.

As suggested by referee one, we are including Muller (2003) as a reference for this in our text.

References cited above:

- Dickson, B., Clack, J. A., Smithson, T. R., Pierce, S. E. 2020. Functional adaptive landscapes predict terrestrial capacity at the origin of limbs. *Nature* 589, 242-245.
- Evans, S. E., Jones, M. E. H. 2010. The Origin, Early History and Diversification of Lepidosauromorph Reptiles. In: Bandyopadhyay, S. (Ed.) *New Aspects of Mesozoic Biodiversity*. Springer Berlin Heidelberg, pp. 27-44.
- Jones, K. E., Angielczyk, K. D., Polly, P. D., Head, J. J., Fernandez, V., Lungmus, J. K., Tulga, S., Pierce, S. E. 2018. Fossils reveal the complex evolutionary history of the mammalian regionalized spine. *Science* 361, 1249-1252.
- Pierce, S. E., Angielczyk, K. D., Rayfield, E. J. 2008. Patterns of morphospace occupation and mechanical performance in extant crocodylian skulls: A combined geometric morphometric and finite element modeling approach. *J. Morphol.* 269, 840-864.
- Pierce, S. E., Angielczyk, K. D., Rayfield, E. J. 2009. Shape and mechanics in thalattosuchian (Crocodylomorpha) skulls: implications for feeding behaviour and niche partitioning. *J. Anat.* 215, 555-576.
- Simões, T. R., Caldwell, M. W., Pierce, S. E. 2020. Sphenodontian phylogeny and the impact of model choice in Bayesian morphological clock estimates of divergence times and evolutionary rates. *BMC Biol.* 18, 191.